# A molecular sensor to quantify the localization of proteins, DNA and nanoparticles in cells

Laura I. FitzGerald[1,2], Luigi Aurelio[1], Moore Chen [1], Daniel Yuen [1], Joshua J. Rennick [1,2], Bim Graham[1] & Angus P. R. Johnston [1,2 ✉]

Intracellular trafficking governs receptor signaling, pathogenesis, immune responses and fate of nanomedicines. These processes are typically tracked by observing colocalization of fluorescent markers using confocal microscopy. However, this method is low throughput, limited by the resolution of microscopy, and can miss fleeting interactions. To address this, we developed a localization sensor composed of a quenched SNAP-tag substrate (SNAP$_{Switch}$) that can be conjugated to biomolecules using click chemistry. SNAP$_{Switch}$ enables quantitative detection of trafficking to locations of interest within live cells using flow cytometry. Using SNAP$_{Switch}$, we followed the trafficking of DNA complexes from endosomes into the cytosol and nucleus. We show that antibodies against the transferrin or hyaluronan receptor are initially sorted into different compartments following endocytosis. In addition, we can resolve which side of the cellular membrane material was located. These results demonstrate SNAP$_{Switch}$ is a high-throughput and broadly applicable tool to quantitatively track localization of materials in cells.

---

[1] Monash Institute of Pharmaceutical Sciences, Monash University, Parkville, VIC, Australia. [2] ARC Centre of Excellence in Convergent Bio-Nano Science and Technology, Monash University, Parkville, VIC, Australia. ✉email: angus.johnston@monash.edu

Determining where material is trafficked following endocytosis is essential for understanding many cellular processes. The intracellular transport of inbound biomolecules is an integral process in cell signalling[1], immune responses[2] and in the trafficking of infectious agents, such as viruses[3] and bacterial toxins[4]. Furthermore, the destination of internalized material is of critical importance for the subcellular delivery of therapeutics[5]. Following uptake, many materials become trapped in endo/lysosomes, preventing access to their site of action[6,7]. For example, nucleic acids for gene silencing[8] or delivery[9,10] require transfer to the cytosol or nucleus, respectively. To design carriers that can efficiently deliver molecules to these locations, we need to understand the trafficking of these vehicles and their ultimate subcellular fate.

The intracellular trafficking of materials is commonly investigated through colocalization analysis. The cargo and intracellular compartments of interest are labelled with fluorescent markers, such as synthetic organic dyes or fluorescent fusion proteins. The distribution of these fluorescent markers is measured by microscopy and statistical analysis (e.g. Pearson's correlation coefficient) quantifies the degree of colocalization[11]. In certain situations, the interpretation of these results is straightforward, such as when complete colocalization (or alternatively, a complete lack of colocalization) occurs. However, the meaningfulness of intermediate coefficient values between these extremes can be difficult to interpret[12]. In addition, traditional colocalization analysis is low throughput, as the number of cells that can be analysed is limited (typically ten to hundreds of cells over hours) compared to other techniques such as flow cytometry (thousands of cells per second). Furthermore, the temporal resolution of current techniques means short-lived events, such as trafficking through a location may be missed. Spatial resolution is also limited as even super-resolution techniques are limited to >40 nm resolution, which is insufficient to determine which side of a subcellular membrane the material is located.

Fluorescent fusion proteins have long been used to label specific locations within cells and to track localization of inbound proteins of interest[13]. More recently, genetically encodable tagging systems have attracted attention as an alternative. These techniques, including the biarsenical-tetracysteine tag[14], VIPER[15], HaloTag[16], CLIP-tag[17] and SNAP-tag[18], were developed to allow fluorescent labelling with small organic dyes, which can be superior in terms of brightness and photostability[19]. Of these, one of the most established is the SNAP-tag, which uses an engineered variant of the 19 kDa O[6]-alkylguanine-DNA alkyltransferase repair enzyme that reacts with benzylguanine, covalently linking a fluorescent label to the protein.

Sensors with triggerable fluorescence are powerful tools for studying interactions of materials with cells, as they often have improved signal-to-noise ratios. Some examples include exploiting a change in nucleic acid hybridization[20,21] or the interaction of a substrate with an enzyme[22] to generate a fluorescent signal. Advances to the SNAP-tag system have been made by using quenched dyes, which allows live cell, wash-free imaging[23,24]. However, the quenched substrates are engineered to be membrane permeable and are not designed for attachment to DNA, proteins or nanoparticles, which prevents their use in studying the localization of internalized material.

To overcome the limitations with conventional colocalization analysis, we have developed a new quenched SNAP-tag substrate (SNAP$_{Switch}$) that can be conjugated to biomolecules to investigate their localization in live cells. The sensor is attached to a material of interest using click chemistry and remains non-fluorescent until it reaches a SNAP-tagged location of interest within the cell (Fig. 1). This allows cargo to be detected only when it reaches the location of interest. This is a significant advantage over traditional fluorescent tags, as high-throughput techniques such as flow cytometry can be used to quantify localization, without the need for microscopy- and image-based colocalization analysis. We have demonstrated the stability and activation of the sensor both in solution and in vitro. We then applied this method to probe the differences in trafficking of antibodies bound to cell-surface receptors. We also followed the journey of DNA delivered with a transfection reagent (Lipofectamine 3000) from endosomes, into the cytosol and finally its delivery to the nucleus.

## Results

**Design of the SNAP$_{Switch}$ localization sensor**. SNAP$_{Switch}$, is based on benzylguanine, the native substrate for the SNAP-tag. A fluorophore and azido lysine (for subsequent click conjugation to the material of interest) were coupled to the guanine group of benzylguanine, and a quencher was coupled to the benzyl group (Fig. 1b). The sensor is engineered so that when SNAP$_{Switch}$ interacts with SNAP-tag, the quencher (QSY-21) is transferred to the SNAP-tag, while a fluorophore (Cy5) becomes fluorescent and remains attached to the protein/DNA/nanoparticle. After activation, the fluorescence is permanently switched on and the sensor remains attached to the material of interest. These features of SNAP$_{Switch}$ offer significant advantages over typical image-based colocalization analysis and other assays for endosomal escape such as split green fluorescent protein (GFP)[25]. Signal from SNAP$_{Switch}$ accumulates over time as more interactions occur enabling quantification of material transitioning through specific locations of the cell, such as cargo passing through an endosome into the cytosol. The fluorophore is pH insensitive within a range relevant to the endocytic pathway (pH 4–10)[26], while most GFP variants are not[27], which is an issue where endocytic processes are involved due to acidification endosomes. Finally, the labelled material is not anchored to the SNAP-tag after an interaction that could potentially block further trafficking. This means subsequent trafficking of the material can be observed either in the same sample or at different time points, using flow cytometry to avoid photobleaching.

**Sensor characterization**. Following synthesis, we evaluated the spectral properties of SNAP$_{Switch}$. The sensor absorbed strongly ~658 nm (Supplementary Fig. 1a), between the maximum absorption wavelengths of Cy5 and QSY-21. SNAP$_{Switch}$ exhibited low fluorescence in solution, which drastically increased following treatment with SNAP-tag (Supplementary Fig. 1b). This change in intensity was also observed via in-gel fluorescence (Fig. 2a, b). The fluorescent band at <10 kDa corresponds to the cleaved free dye, and not the molecular weight of the SNAP-tag, as the quencher is transferred to the SNAP-tag and the fluorophore is released. Labelling of free SNAP-tag with the commercial substrate SNAP-Cell SiR 647 resulted in a fluorescent band at ~20 kDa, while unreacted fluorescent substrate appeared at a similar position to the cleaved SNAP$_{Switch}$ (Fig. 2a).

SNAP$_{Switch}$ was stable in the quenched state, showing no increase in fluorescence intensity over a 90-min incubation period when no SNAP-tag was present. However, the fluorescent signal rapidly activated in the presence of SNAP-tag. Incubation of SNAP$_{Switch}$ with 20 equivalents of SNAP-tag resulted in an ~10-fold increase in Cy5 fluorescence intensity after 90 min (Fig. 2c). Activation of the sensor was rapid, with the second-order rate constant estimated to be $111 \pm 5$ $M^{-1}$ $s^{-1}$ (Supplementary Fig. 2). This is significantly faster than most bioorthogonal click reactions (e.g. strain-promoted alkyne-azide cycloadditions), which typically range from 0.1 to 10 $M^{-1}$ $s^{-1}$ [28], and is similar to the activation of non-attachable quenched SNAP-tag substrates[24,29].

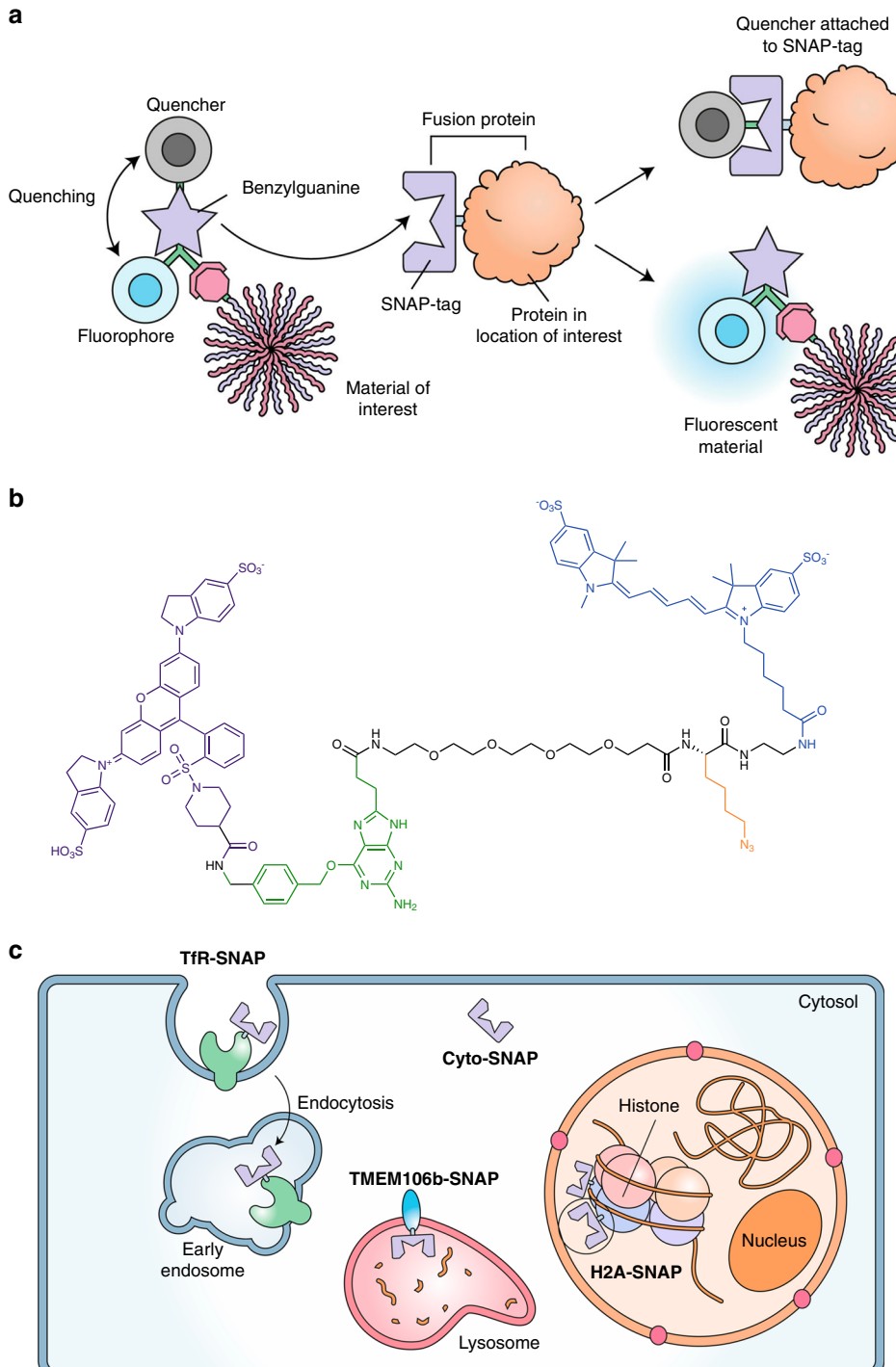

**Fig. 1 Design of the SNAP$_{Switch}$ localization sensor.** SNAP$_{Switch}$ localization sensor can be used to quantify trafficking of proteins, DNA or nanoparticles to a subcellular compartment. **a** SNAP$_{Swtich}$ scheme. SNAP$_{Switch}$ consists of a quencher, fluorophore and clickable group for attachment conjugated to either side of a benzylguanine group. When SNAP$_{Switch}$-labelled material reaches a cellular compartment labelled with a SNAP-tag (e.g. early endosome, cytosol or nucleus), the quencher is transferred to the SNAP-tag, breaking the quenching interaction and allowing the sensor to fluoresce. **b** SNAP$_{Switch}$ chemical structure. QSY-21 (purple) is conjugated to the side of the benzylguanine substrate (green) that is transferred to the SNAP-tag. The substrate is linked to an azide (orange) for attachment to the particle or protein of interest through a PEG linker. The fluorophore Cy5 (blue) resides on the side of the substrate that remains with the material after interaction with the SNAP-tag. **c** SNAP-tag can be localized to different areas of the cell to probe trafficking to different cellular compartments. SNAP-tag fused to the transferrin receptor to probe early endosome localization, to transmembrane protein 106b for late endo/lysosomes trafficking, in the cytosol for endosomal escape of material following endocytosis and in the nucleus for transport of nucleic acids following delivery.

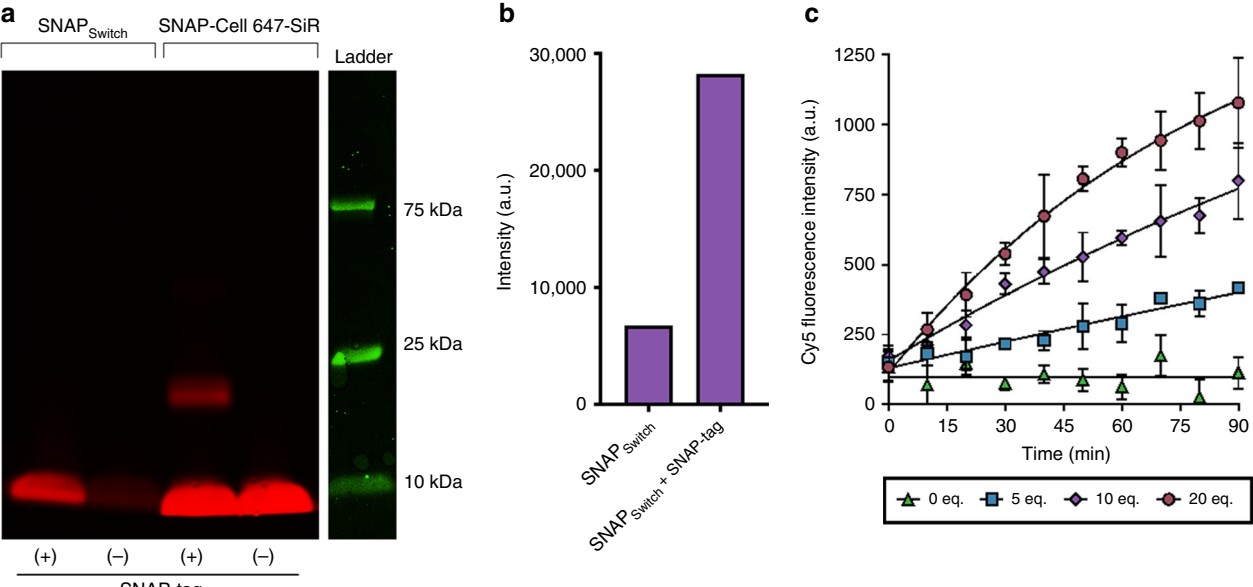

**Fig. 2 Characterization by in-gel fluorescence and in solution. a** In-gel detection of the quenched SNAP-tag substrate of 600 μM SNAP$_{Switch}$ or fluorescent SNAP-Cell 647-SiR with or without 5 μM SNAP-tag protein incubated at 37 °C for 30 min. **b** Densitometric analysis of SNAP$_{Switch}$ activation with and without SNAP-tag incubation. **c** Fluorescence activation of 0.10 μM SNAP$_{Switch}$ in the presence of SNAP-tag protein at 27 °C over 1.5 h. The mean of three wells is plotted with error bars representing the standard deviation ($n = 3$). The data are fitted with an exponential one-phase association curve.

To demonstrate SNAP$_{Switch}$ has improved sensitivity compared to other fluorescence assays, we investigated the signal generated from a split-GFP complementation system, which has previously been used to detect cytosolic delivery of cell penetrating peptides[25,30,31]. Here, GFP is split into a large (GFP$_{1-10}$) and small (GFP$_{11}$) fragment, rendering it almost non-fluorescent until an interaction occurs between the two fragments. With 10 equivalents of protein (SNAP-tag or GFP$_{1-10}$) to its interaction partner (SNAP$_{Switch}$ or GFP$_{11}$), a significant change above background was seen within 10 min ($p < 0.05$, unpaired $t$ test) for SNAP$_{Switch}$ (Fig. 2c), while split GFP took 60 min for a significant signal to be detected ($p < 0.05$, unpaired $t$ test) (Supplementary Fig. 3). After 90 min, the SNAP$_{Switch}$ signal was 7-fold higher than the control, while split GFP increases was only 1.5-fold higher.

**In vitro activation of SNAP$_{Switch}$.** Having established that SNAP$_{Switch}$ was quenched efficiently and was responsive to SNAP-tag in solution, we moved to demonstrate its response in vitro. Activation of SNAP$_{Switch}$ by SNAP-tag expressed in cells was tested by fusing SNAP-tag to the human transferrin receptor (hTfR-SNAP). hTfR-SNAP was stably introduced into NIH/3T3 (3T3) cells using lentiviral transduction and was expected to localize with endogenous mouse TfR. SNAP$_{Switch}$ was then conjugated to either anti-mouse TfR (anti-mTfR) or anti-human TfR (anti-hTfR) antibodies. In contrast to holotransferrin, which rapidly recycles back to the surface, these antibodies are expected to be trafficked to the lysosome over time[32].

To determine if SNAP$_{Switch}$ could be activated non-specifically on the cell surface, we also conjugated the sensor to an anti-CD44 antibody. CD44 is a membrane glycoprotein expressed by 3T3 cells[33] that is involved in cell adhesion, signalling and is predominately located on the plasma membrane[34]. Binding of antibodies at low temperature (4 °C) was avoided as anti-CD44 is internalized by the clathrin-independent carrier/glycosylphosphatidylinositol-anchored protein-enriched early endosomal compartment (CLIC/GEEC) pathway[35], which takes longer to recover than clathrin-mediated endocytosis on return to 37 °C[36].

An additional fluorophore, BODIPY FL (BDP-FL) was also attached to the antibodies to detect the proteins, while the sensor was switched off. This also allowed us to account for slight variations in the amount of antibody association by calculating the ratio of the Cy5 signal generated by the SNAP$_{Switch}$ to the amount of antibody present determined by the BDP-FL signal.

SNAP$_{Switch}$ was specifically activated by proteins that colocalized with hTfR-SNAP. As expected, anti-CD44 and anti-mTfR associated with both wild-type and hTfR-SNAP-expressing 3T3 cells (Fig. 3a). Anti-hTfR bound only to cells expressing the hTfR-SNAP fusion protein, confirming the specificity of anti-hTfR for hTfR. SNAP$_{Switch}$ was not activated in cells lacking the SNAP-tag, as no Cy5 signal was observed for any antibody in wild-type 3T3 cells after 1 h (Fig. 3b), and no increase in SNAP$_{Switch}$ signal was observed over 4 h (Fig. 3c). This demonstrates the in vitro stability and specificity of the sensor. In cells that express hTfR-SNAP, SNAP$_{Switch}$ was activated when conjugated to anti-mTfR (~400 a.u.) and anti-hTfR (~900 a.u.) (Fig. 3b). Activation occurred rapidly, with the majority of signal generated in the first hour, and only a slight increase thereafter (Fig. 3d). Activation of SNAP$_{Switch}$ attached to anti-mTfR by SNAP-hTfR shows that the two isotypes of the receptor colocalize, and that their proximity is such that the SNAP-tag has access to the sensor. This was confirmed by colocalization of fluorescently labelled anti-hTfR and anti-mTfR (Supplementary Fig. 4). Anti-mTfR and anti-hTfR also colocalized with hTfR-SNAP labelled with a cell-permeable fluorescent SNAP substrate (Fig. 3e, f).

In contrast to the anti-TfR antibodies, minimal activation of SNAP$_{Switch}$ attached to anti-CD44 was observed (~30 a.u.) over 1 h (Fig. 3b), even though there was significant association with cells (Fig. 3a). This suggests that anti-CD44 was not trafficked to compartments containing the hTfR-SNAP fusion over this time period. This was corroborated by fluorescence microscopy, which demonstrated that the majority of anti-CD44 remained bound to the cell surface and did not colocalize with anti-hTfR or anti-mTfR after 1 h (Supplementary Fig. 5). Limited activation was detected after 4 h (Fig. 3d), indicating that small amounts of anti-CD44 may eventually reach endocytic vesicles bearing TfR-SNAP.

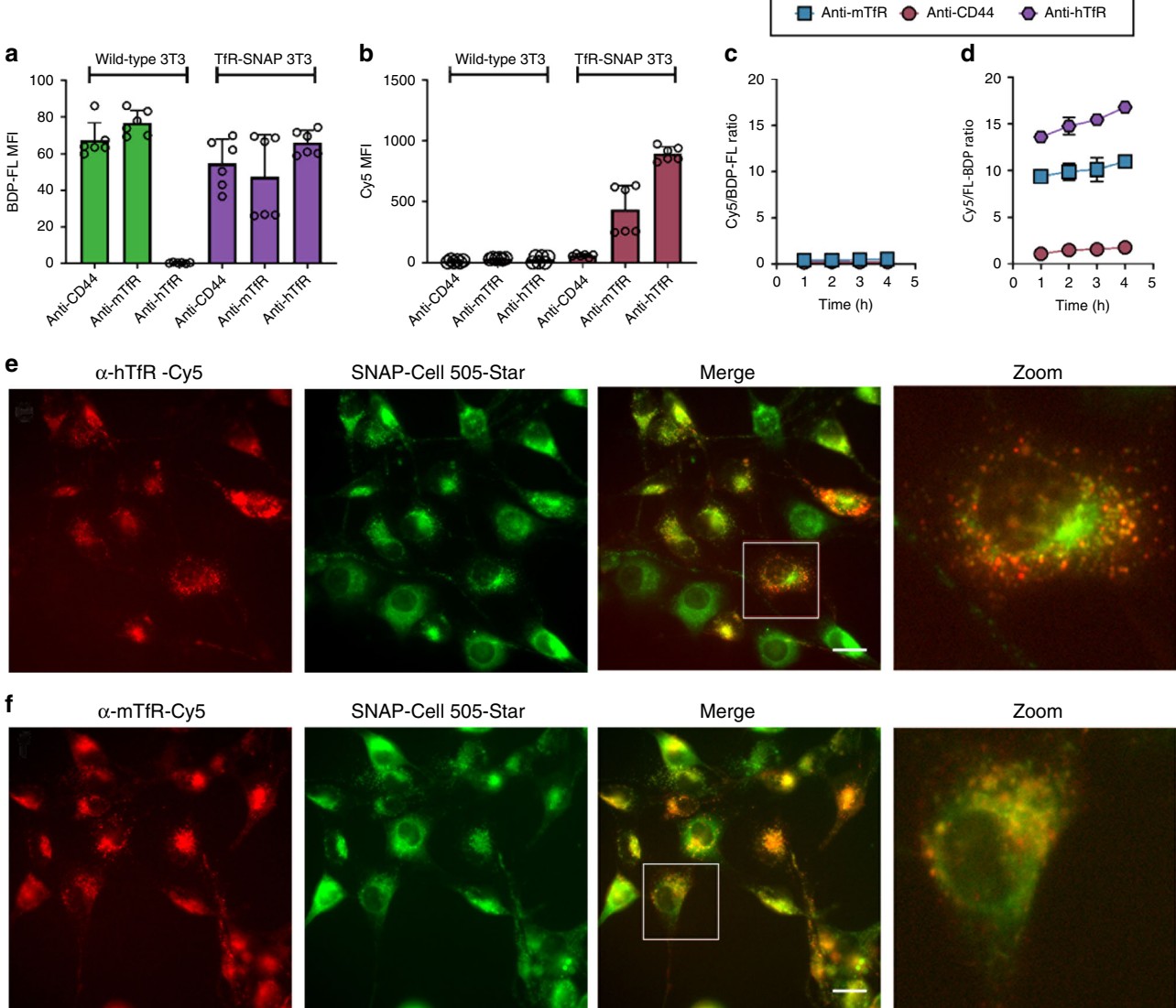

**Fig. 3 In vitro activation of SNAP_Switch.** SNAP_Switch conjugated to anti-transferrin antibodies is activated by SNAP-tag fused to the transferrin receptor (TfR-SNAP). **a** Association of BDP-FL-labelled anti-CD44, anti-mTfR (anti-mouse TfR) and anti-hTfR (anti-human TfR) at 1 h in 3T3 or 3T3 cells stably expressing SNAP-tag fused to the human TfR (TfR-SNAP), measured by flow cytometry. **b** Activation of SNAP_Switch on antibodies in wild-type 3T3 or TfR-SNAP cells after 1 h. SNAP_Switch activation on antibodies over time in **c** wild type or **d** TfR-SNAP cells, measured by the ratio of Cy5 to BDP-FL mean fluorescence intensity via flow cytometry at each time point. The mean fluorescence intensity or ratio is plotted with error bars representing the standard deviation of two independent experiments performed in triplicate ($n = 6$). Fluorescence microscopy images of Cy5-labelled (**e**) Cy5-labelled (red) anti-hTfR and (**f**) anti-mTfR in TfR-SNAP cells with the SNAP-tag stained with SNAP-Cell 505-Star (green). Pearson's correlation coefficient: anti-hTfR = 0.760, SD = 0.045, anti-mTfR = 0.662, $n = 5$. Scale bar = 20 μm. Source data are presented in a Source Data file.

This was confirmed by fluorescence microscopy, with punctate structures containing both anti-CD44 and anti-hTfR antibody observed after 4 h (Supplementary Fig. 6). Activation of SNAP_Switch is not a direct result of SNAP_Switch-labelled antibody binding to a SNAP-tagged receptor, but occurs when SNAP_Switch-labelled antibody comes into close proximity with a SNAP-tagged receptor. The SNAP-tagged receptor that activates the SNAP_Switch will likely not be the receptor that the antibody binds to, as the distance between the SNAP_Switch and SNAP-tag will be too great.

**Quantification of SNAP_Switch interactions.** The total amount of antibody that interacts with hTfR-SNAP in live cells can be quantified by ratioing the activated signal observed in cells to the maximal signal following in vitro SNAP_Switch activation.

Treating SNAP_Switch antibody conjugates (also labelled with AF488) with a 30 M excess of SNAP-tag for 1 h fully activates the SNAP_Switch (Supplementary Fig. 7, Fig. 4a) and does not affect association of the antibodies with either wild-type 3T3 or hTfR-SNAP cells (Fig. 4b). Complete pre-activation of the sensor was confirmed as the SNAP_Switch−AF488 ratio for anti-mTfR treated with SNAP-tag was the same in both wild-type ( ~ 6.2) and hTfR-SNAP-tag cells (~6.3) (Fig. 4c).

To quantify localization, the ratio of the signal from SNAP_Switch activated by hTfR-SNAP in live cells to the signal from pre-activated SNAP_Switch was calculated (Fig. 4d). We observed >87% activation of SNAP_Switch attached to anti-huTfR, while only ~32% activation of SNAP_Switch attached to anti-mTfR was observed. This indicates that only a portion of anti-mTfR antibody came into close proximity to the hTfR.

**Resolving membrane orientation of cargo.** Understanding which side of a membrane an antibody or therapeutic cargo is

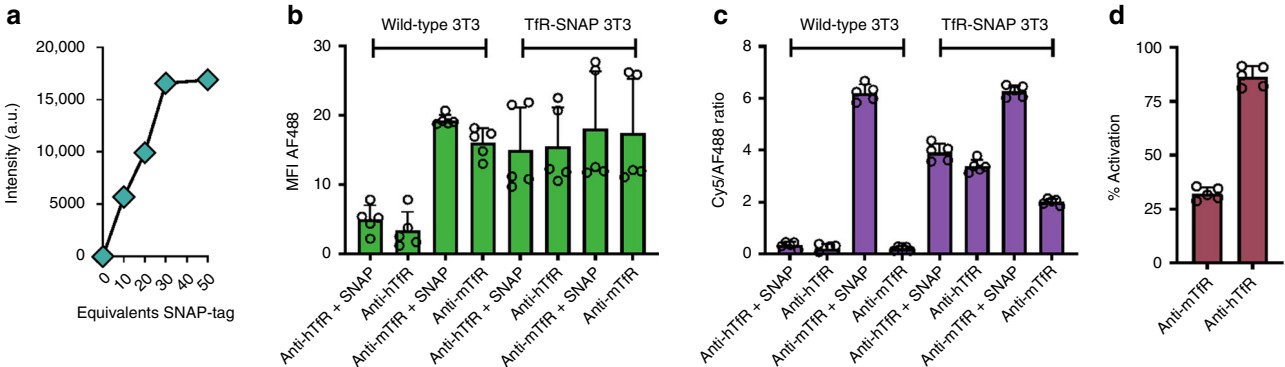

**Fig. 4 Quantification of SNAP$_{Switch}$ interactions. a** Densitometric analysis of SNAP$_{Switch}$ activation, in-gel conjugated to anti-mTfR and treated with 0–50 equivalents of SNAP-tag for 1 h at 37 °C, relative to the degree of labelling. Anti-mTfR dual-labelled with AF488 and SNAP$_{Switch}$ incubated for 1 h at 37 °C with or without 30 equivalents of SNAP-tag before incubation in either 3T3 or TfR-SNAP cells and analysed by flow cytometry. **b** Association measured by the AF488 signal. **c** SNAP$_{Switch}$ activation determined by the Cy5/AF488 ratio and **d** the percentage of SNAP$_{Switch}$ activated on anti-mTfR in TfR-SNAP cells. The mean fluorescence intensity, ratio or percentage activation is plotted with error bars representing the standard deviation of two independent experiments performed in triplicate and duplicate ($n = 5$). Source data are presented in a Source Data file.

delivered is crucial for maximizing therapeutic activity. It is of significant interest to deliver material into the cytosol or nucleus in particular. However, cargo often remains trapped on the luminal side of the endosomal membrane and is unable to exhibit cytosolic or nuclear activity. Therefore, it is important to distinguish which side of the membrane the cargo resides on. The thickness of a typical bilayer is ~4 nm[37] and neither live or fixed super-resolution fluorescence imaging techniques currently have sufficient resolution to resolve which side of a membrane a material resides (currently limited to >20 nm[38]). To demonstrate SNAP$_{Switch}$ is able to resolve which side of an endosomal membrane material is delivered to, we fused SNAP-tag to either the N or C terminus of a resident late endo/lysosomal transmembrane protein (TMEM106b). The N terminus of the native protein resides in the cytosol (SNAP-TMEM106b), while the C terminus is located in the endo/lysosomal lumen (TMEM106b-SNAP)[39]. This was confirmed by treating cells stably transfected with TMEM106b-SNAP or SNAP-TMEM106b with SNAP-Surface 647. SNAP-Surface 647 is a membrane-impermeable dye that is not able to diffuse into the cytosol. TMEM106b-SNAP cells (endo/lyosomal lumen SNAP) have strong, punctate fluorescence (Fig. 5a), whereas SNAP-TMEM106b (cytosolic SNAP (Cyto-SNAP); Fig. 5b) cells have limited signal. SNAP$_{Switch}$ conjugated to anti-mTfR was activated in 3T3 cells stably expressing the luminal TMEM106b-SNAP-tag, but no activation of the Cyto-SNAP-TMEM106b was detected (Fig. 5c, d). Association of the antibody with both cell lines was the same (Fig. 5c), but a large increase in SNAP$_{Swtich}$ signal was observed in the C-terminal SNAP-tag cells (Fig. 5d). This shows that as expected for an antibody known to traffic through to lysosomes, the antibody remains contained within the endo/lysosome and does not escape into the cytosol. It also demonstrates the high spatial resolution of the SNAP$_{Switch}$ sensor, which can be achieved using flow cytometry and without the need for super-resolution imaging.

**Uptake, trafficking and delivery of oligonucleotides.** Delivery of nucleic acids to the cytosol and nucleus is critical for gene therapy applications, but the efficiency of this transport is difficult to determine. Lipofectamine 3000, a cationic lipid formulation that condenses DNA into nano-sized complexes[40], is widely used as a transfection reagent that induces endosomal escape. We explored the ability of SNAP$_{Switch}$ to track cargo through the endocytic pathway to other locations of interest within the cell.

We used a fusion to histone protein H2A and the fluorescent protein mTurquoise (H2A-mTurq-SNAP) to install SNAP-tag in the nucleus of transiently transfected cells (Supplementary Fig. 8), with expression found to be sustained for at least 16 h (Supplementary Fig. 9). The expression of SNAP-tag in the mTurquoise-positive population was confirmed by the high SNAP-Cell SiR 647 signal (>8000 a.u.) within these cells (Fig. 6a).

Delivery of DNA cargo to the nucleus was evaluated using flow cytometry by measuring the SNAP$_{Switch}$ signal in live cells positive for mTurquoise. To probe the delivery of DNA to the nucleus, dual-labelled Lipofectamine/DNA complexes were formulated using a 20-mer oligonucleotide sequence labelled with AF488 and SNAP$_{Switch}$. Signal from SNAP$_{Switch}$ was expected to be generated from oligonucleotides that have dissociated from Lipofectamine in addition to that still complexed, but near SNAP-tag. The SNAP$_{Switch}$ signal in H2A-mTurq-SNAP cells was not significantly higher than in wild-type cells over the first 2 h, suggesting no exposure of the delivered DNA to nuclear components during this period. From 2 to 4 h, the SNAP$_{Switch}$ signal increased slightly, followed by a steady increase in intensity up to 16 h. No difference in association of the complexes with wild-type or H2A-mTurq-SNAP HEK cells was observed (Fig. 6b), allowing for direct analysis of the SNAP$_{Switch}$ signal to determine nuclear access of the delivered DNA (Fig. 6c). Background signal from the SNAP-negative cells showed a small increase in signal over 16 h, which was subtracted from the Cy5 signal from the mTurquoise-positive cells. This shows that, as expected, there is a lag time between the Lipofectamine/DNA complex being added to the cells and arrival of the DNA in the nucleus. The increase in signal was not due to cytotoxic effects of Lipofectamine, as cells transfected at different time points and treated with a viability stain had similar viability (mean fluorescence intensity (MFI) ~11) to non-transfected cells (MFI ~8.4), while the signal from apoptotic cells was ~100-fold higher (MFI ~1000) (Supplementary Fig. 10). It is possible that the DNA cargo comes in contact with the SNAP-tagged H2A during mitosis (when the nuclear membrane brakes down) and that direct delivery of DNA through the nuclear membrane does not occur[41]. Regardless of the mechanism, SNAP$_{Switch}$ is able to determine if and when the delivered DNA can access the nuclear components of the cell.

To further probe the trafficking of the Lipofectamine/DNA complexes in other compartments, we also engineered cells to express a Cyto-SNAP to measure endosomal escape. Fluorescence

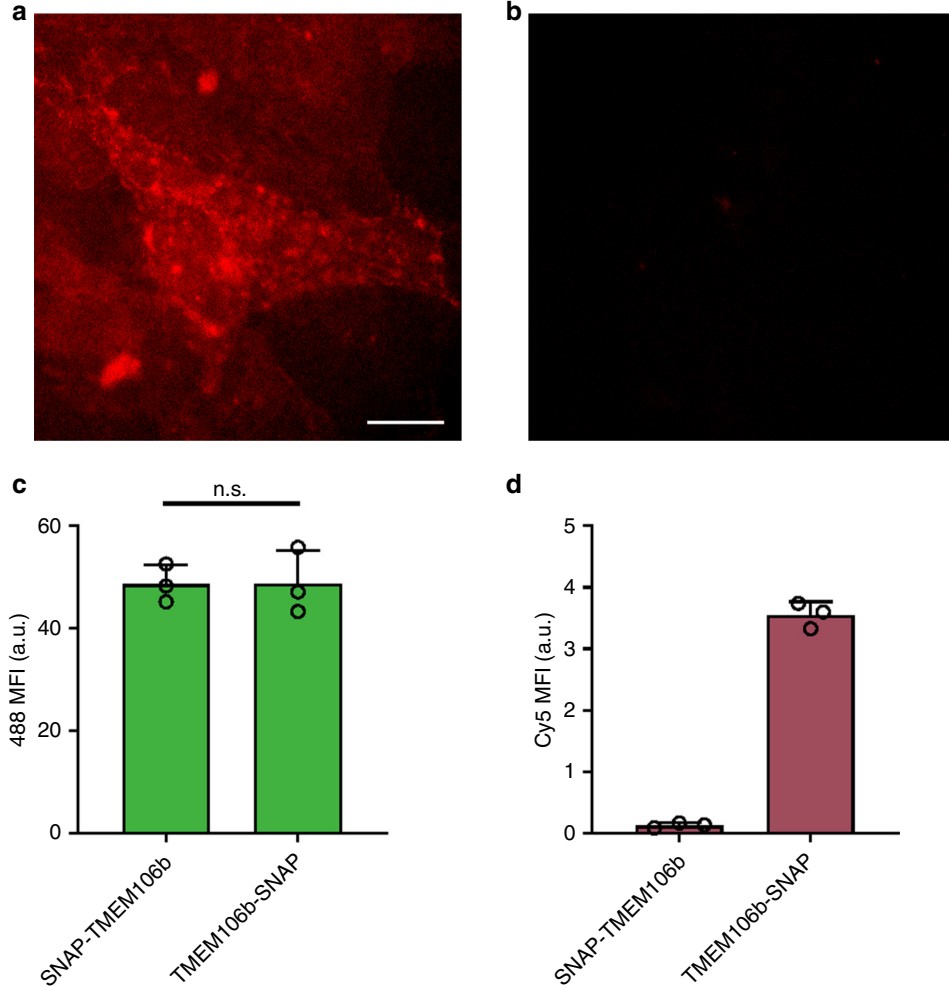

**Fig. 5 Resolving membrane orientation of cargo.** Fluorescence microscopy images of 3T3 cells stably expressing SNAP-tag fused to the **a** C terminus (luminal, SNAP-TMEM106b) or **b** N terminus (cytosolic, TMEM106b-SNAP) of transmembrane protein 106b in late endo/lysosomes, treated with SNAP Surface Alexa Fluor 647. Scale bar = 10 µm. SNAP$_{Switch}$ conjugated to anti-mTfR is activated by SNAP-tag on the luminal side of endo/lysosomal membrane only. Cells incubated with 5 µg mL$^{-1}$ anti-mTfR dual labelled with BDP-FL and SNAP$_{Switch}$ or without (control) for 1 h before analysis by flow cytometry. The average of cells expressing SNAP-TMEM106b or TMEM106b-SNAP and without antibody was subtracted as background from each sample with antibody treatment. The mean fluorescence intensity of the **c** BDP-FL or **d** SNAP$_{Swtich}$ signal from flow cytometry is plotted with error bars representing the standard deviation of one experiment in triplicate (n.s. non-significant, $P = 0.9871$, two-sided, unpaired $t$ test, $n = 3$). Source data are presented in a Source Data file.

microscopy confirmed a diffuse signal throughout the cells for the Cyto-SNAP construct (Supplementary Fig. 11). To confirm Cyto-SNAP localized in the cytosol and not in lysosomes, we looked for activation of SNAP$_{Switch}$ conjugated to anti-mTfR. We[32], and others, have previously shown anti-TfR antibodies localize to endo/lysosomes[42]. SNAP$_{Switch}$-labelled anti-mTfR showed no activation in Cyto-SNAP-transfected cells after 4 h, but induced a robust response in hTfR-SNAP cells (Supplementary Fig. 12). This demonstrates that Cyto-SNAP does not localize to the lysosomes.

We then used the SNAP$_{Switch}$ sensor to visualize both cytosolic delivery and nuclear exposure by fluorescence microscopy. Cyto-SNAP or H2A-SNAP cells were incubated with SNAP$_{Switch}$ Lipofectamine/DNA complexes for 16 h. The response from the sensor was localized to different regions of cells expressing Cyto-SNAP or H2A-SNAP. In Cyto-SNAP cells, SNAP$_{Switch}$ signal was observed as a combination of bright punctate structures as well as disperse cellular fluorescence (Fig. 6d). As Cyto-SNAP does not localize to the lysosomes, the punctate structures suggest that a proportion of the DNA may still be complexed with

Lipofectamine 3000 in the cytosol or is associated with the membrane remnants of ruptured endo/lysosomes. The fluorescent signal from H2A-SNAP cells was confined to the nucleus (Fig. 6e). The reduced amount of punctate structures suggests that the DNA had dissociated from Lipofectamine either before or after contact with H2A. No signal was observed in wild-type cells incubated with SNAP$_{Switch}$ Lipofectamine/DNA complexes and transfected with an empty plasmid (Fig. 6f), demonstrating limited non-specific activation and that SNAP$_{Switch}$ is only activated when it comes in close proximity to a SNAP-tag.

As with most microscopy data used to determine localization, it is difficult to know if these cells demonstrating escape were fated for transfection or apoptosis. However, the compatibility of SNAP$_{Switch}$ with flow cytometry allows for the analysis of thousands of cells, limiting the chance of data being skewed by analysing cells that would undergo apoptosis.

To demonstrate the high-throughput potential of SNAP$_{Switch}$ using flow cytometry, we further probed the trafficking of the Lipofectamine/DNA complexes over time. Cell lines stably expressing TfR-SNAP, Cyto-SNAP, and H2A-SNAP were

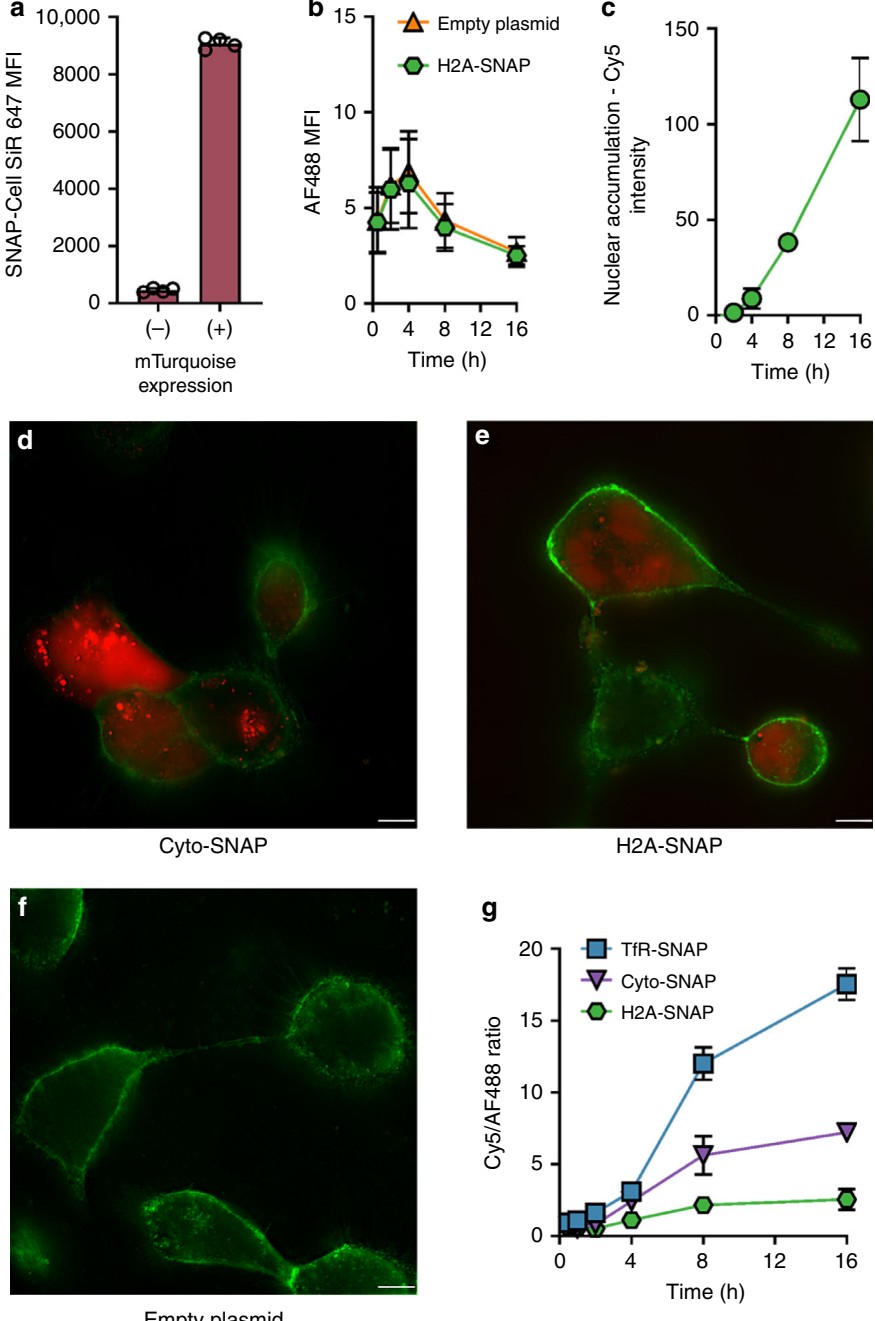

**Fig. 6 Uptake, trafficking and delivery of oligonucleotide.** HEK293 cells were transfected to transiently express H2A-mTurquoise-SNAP-tag and analysed by flow cytometry. **a** SNAP-Cell SiR 647 signal in cells gated positive or negative for mTurquoise expression. **b** Association of Lipofectamine 3000 complexes with cells transfected with H2A-SNAP or an empty plasmid. **c** SNAP$_{Switch}$ signal in cells positive for mTurquoise expression with the signal from negative cells subtracted as background. Deconvolved fluorescence microscopy images of oligonucleotide labelled with SNAP$_{Switch}$ and incubated with HEK293 cells for 16 h, transfected with **d** Cyto-SNAP, **e** H2A-SNAP and **f** an empty plasmid and surface stained with wheat germ agglutinin Alexa Fluor 488, scale bar = 10 μm. **g** HEK cells stably expressing TfR-SNAP, Cyto-SNAP or H2A-SNAP and transfected with Lipofectamine 3000 complexes containing AF488 and SNAP$_{Switch}$ oligonucleotides over time, analysed by flow cytometry. The mean fluorescence intensity (**a–c**) or ratio (**g**) is plotted with error bars representing the standard deviation, from two experiments in duplicate (**a–c**, $n = 4$) or one experiment in triplicate (**g**, $n = 3$). Source data are presented in a Source Data file.

generated using lentiviral transduction of HEK293 cells. The behaviour of the dual-labelled oligonucleotide Lipofectamine 3000 complexes was then followed over 16 h. The association of the complexes across the cell lines was similar (Supplementary Figs. 13a and 14a). Again, to account for the slight variations in cell association, the ratio of SNAP$_{Switch}$ to AF488 signal at each time point was calculated. The SNAP$_{Switch}$ signal for TfR-SNAP

cells showed a small increase in intensity over the first 2 h, whereas the Cyto-SNAP and H2A-SNAP showed minimal activation (Fig. 6g, Supplementary Figs. 13b and 14b). From 4 to 16 h the TfR-SNAP cells showed significant activation, indicating that lipoplexes are trafficked into TfR-positive vesicles. As expected, the signal from the Cyto-SNAP cells was significantly lower than the TfR-SNAP cells after 16 h, indicating

that lipoplexes get trapped in TfR-positive vesicles and the endosomal translocation is relatively inefficient. Finally, the H2A-SNAP cells showed the lowest activation, indicating that efficiency of transport from the cytosol to nuclear constituents is also inefficient.

## Discussion

The uptake and intracellular trafficking of materials is important for regular cellular function, the mechanisms behind disease pathogenesis and drug delivery. However, current techniques to determine localization are limited by low throughput, are open to subjective interpretation and can miss momentary interactions. Through synthesis of a quenched and attachable SNAP-tag substrate, we have developed a sensor that enables high throughput (thousands of cells per sample, measured in less than a minute) and quantitative tracking of biomacromolecules in live cells. SNAP$_{Switch}$ permanently switches on when labelled material encounters SNAP-tagged proteins in specific subcellular locations. This is a significant improvement over other systems, such as split-GFP[25], where both parts must remain bound for the signal to occur, potentially blocking subsequent protein trafficking. It is also a significant improvement over image-based colocalization assays, which struggle to detect transitionary processes such as those that occur in endosomal trafficking and escape. The use of flow cytometry means that robust statistical analysis of >10,000 cells can be performed, rather than performing image-based colocalization on <100 cells.

Using SNAP$_{Switch}$, we have demonstrated the ability to quantify trafficking of cell-surface receptors, such as TfR and CD44. We show that while antibody against CD44 mostly localizes to the cell membrane, small amounts do meet TfR-positive compartments over time. The lack of initial SNAP$_{Switch}$ activation attached to anti-CD44 highlights two points: first, even though both hTfR-SNAP and anti-CD44 are present on the surface of the cell, the SNAP$_{Switch}$ is not activated. This indicates that hTfR-SNAP and anti-CD44 bound to CD44 do not come close enough on the surface of the cell for an interaction to occur. Second, the trafficking pathway followed by anti-CD44 is different to that of the two anti-TfR antibodies. CD44 has been reported to internalize through CLIC/GEEC endocytosis, rather than the clathrin-dependent pathway followed by the transferrin receptor. The partial activation after further incubation suggests that a proportion of anti-CD44 does reach eventually reach the same compartments as the TfR-SNAP fusion.

SNAP$_{Switch}$ is also able to determine localization within organelles at very high resolution. By attaching the SNAP-tag to the cytosolic domain of an endosome (SNAP-TMEM106b), or the luminal domain of an endosome (TMEM106b-SNAP), we were able determine that anti-mTfR was only present on the luminal domain of an endosome. We observed no signal from SNAP-TMEM106b, demonstrating that SNAP$_{Switch}$, can resolve material localization separated by the thickness of an endosomal membrane (<10 nm).

In addition, we also tracked the uptake of DNA complexed with Lipofectamine 3000 into endosomes, followed by its translocation into the cytosol and subsequent trafficking to nuclear components in live cells. We show the amount present in each of these locations decreases along this journey, which supports the current view that endosomal escape is a major bottleneck in the delivery of nucleic acids[43]. A critical limitation of using microscopy to pinpoint endosomal escape is the subjectivity of image analysis. Images are often assigned as having a "punctate" (trapped material) or "diffuse appearance" (escaped material)[7]. The use of flow cytometry to quantify signal from thousands of cells removes this subjectivity.

SNAP$_{Switch}$ provides a new method to follow the journey of material in live cells to subcellular locations following endocytosis. Unlike endpoint assays, which give a yes or no answer, for if delivery has occurred, the SNAP$_{Switch}$ can pinpoint the point in the trafficking pathway that has failed or is inefficient. The low background fluorescence of the sensor when in its off state and the large increase when the SNAP-tag is encountered provides the basis for a system to definitively identify when internalized cargo has reached a specific location in the cell. As the sensor is only activated in close proximity to the SNAP-tag, co-trafficking of proteins and receptors can be investigated. In addition, the ability to detect delivery to subcellular compartments will provide a useful tool to study the intracellular trafficking, endosomal escape and the subcellular delivery of nanoparticles, proteins and nucleic acids.

## Methods

**Materials**. Peptide synthesis reagents, including 2-chlorotrityl chloride resin, $N(\alpha)$-Fmoc-$N(\varepsilon)$-azide-L-lysine and (benzotriazol-1-yloxy)tripyrrolidinophosphonium hexafluorophosphate (PyBOP), were purchased from ChemImpex. Fmoc-$N$-amido-dPEG$_4$-acid was purchased from Quanta Biosciences. Sulfonated cyanine 5 succinimidyl ester (Cy5-NHS) and BDP-FL succinimidyl ester (BDP-FL-NHS) were purchased from Lumiprobe. Solvents and other organic synthesis reagents, including acetonitrile (MeCN), ethylenediamine, dichloromethane (DCM), dimethylformamide (DMF), methanol (MeOH), $N,N$-diisopropylethylamine (DIPEA), anhydrous dimethyl sulfoxide (DMSO), hexafluoroisopropanol (HFIP), triisopropylsilane (TIPS), dithiothreitol (DTT), piperidine and phosphate-buffered saline tablets (PBS), were purchased from Sigma-Aldrich. Sulfonated QSY-21 carboxylic acid (sQSY-21-COOH)[44] and Fmoc-benzylguanine carboxylic acid (Fmoc-BG-COOH)[45] were synthesized and purified in house using published procedures.

Fluorescent SNAP-tag substrates and the SNAP-tag plasmid, pSNAPf, were purchased from New England Biolabs. Additional labels and reagents including Alexa Fluor 488 succinimidyl Ester (AF488-NHS), Alexa Fluor 488 Azide, Wheat Germ Agglutinin Alexa Fluor 488 conjugate, Click-IT Succinimidyl Ester DIBO Alkyne (DIBO-NHS), Zeba Spin Columns 7 K Molecular Weight Cut-Off, Lipofectamine 3000 Transfection Reagent, Hoechst 33342 Trihydrochloride Trihydrate and general tissue culture supplies were obtained from Thermo Fisher Scientific. Zombie Yellow Fixable Viability Kit was purchased from BioLegend.

The monoclonal immunoglobulin G1 (IgG1) anti-mouse G1 (5035-41.1D) was purchased from Novus Biologicals. Purified mouse monoclonal IgG1 anti-hTfR antibody (clone OKT9)[46] and anti-mTfR IgG2a monoclonal antibody (clone TIB-219)[47] were purchased from Antibody Services at the Walter and Eliza Hall Institute Biotechnology Centre.

**Instrumentation**. Liquid chromatography-mass spectrometry (LCMS) was performed on an Agilent 6100 Series Single Quad LCMS with a photodiode array detector (214/254 nm) coupled to an Agilent 1200 Series HPLC with a G1311A quaternary pump, G1329A thermostated autosampler and 1200 Series G1314B variable wavelength detector with a scan range between 100 and 1000 $m/z$ and a 5-min acquisition time.

High-performance liquid chromatography (HPLC) was performed on an Agilent 1260 series modular HPLC fitted with a G1312B binary pump, G1316A compartment equipped with an Agilent Eclipse Plus C18 3.5 μm, 4.6 × 100 mm$^2$ column and a G1312B diode array detector using an elution protocol of 0–10 min, with a water/MeCN 0.1% trifluoroacetic acid (TFA) gradient from 5 to 100% MeCN at a flow rate of 1 mL min$^{-1}$. Preparative HPLC used a Grace Alltima C8 5μ particle size, 22 × 250 mm$^2$ column.

High resolution-mass spectrometry (HRMS) was performed on a Waters LCT TOF LCMS Mass Spectrometer coupled to a 2795 Alliance Separations module. All data were acquired, and the mass was corrected via a dual-spray Leucine Enkephaline reference sample. Mass spectra were created by averaging the scans across each peak and background subtracted of the total ion chromatogram. Acquisition and analysis were performed using the Masslynx software version 4.1.

Ultraviolet (UV)/visible absorbance spectroscopy was performed with a NanoDrop Spectrophotometer ND-1000. Fluorescence spectra in solution were collected with a Shizmazdu RF-5310PC fluorescence spectrophotometer with a HORIBA Synchronous Microsense cuvette.

Gel fluorescence images were obtained on an Amersham Typhoon 5 Biomolecular Imager (GE Healthcare Life Sciences) and analysed in ImageJ (version 1.52g, Wayne Rasband, National Institute of Health, USA). Fluorescence intensity measurements in solution were performed with a PerkinElmer EnSpire Multilabel plate reader operating at 27 °C. All absorbance measurements were obtained with a NanoDrop ND-1000.

**Cell culture**. 3T3 (ATCC: CRL-1658) and HEK293A (HEK, Thermo Fisher Scientific R70507) were maintained in Dulbecco's modified Eagle's medium (DMEM),

high glucose (GlutaMAX) with phenol red and 20% (3T3) or 10% (HEK293A) foetal bovine serum (FBS) and 1% penicillin/streptomycin at 37 °C with 5% $CO_2$.

**Flow cytometry**. Ten thousand events per sample were analysed with a Stratedigm S1000EXI flow cytometer (Stratedigm, California, USA) using 405 nm excitation with emission collected between 565 and 595 nm (for Zombie Yellow), 405 and 552 nm excitation with emission collected between 415 and 475 nm (for mTurquoise), 488 nm excitation with emission collected between 515 and 545 nm (for Alexa Fluor 488/BDP-FL) and 642 nm excitation with emission collected between 661 and 690 nm (for Cy5/SNAP-Cell SiR 647). FCS3.0 files were exported using CellCapTure Analysis Software (version 4.0 RC4, Stratedigm, California, USA) and imported into FlowJo (version 8, Tree Star, Oregon, USA).

Following gating (Supplementary Figs. 15 and 16), the geometric mean fluorescence intensity of untreated cells was taken as the background. The average of this was then subtracted from each sample. These values were then plotted directly or the ratio of Cy5 to BDP-FL or AF488 in each well was calculated. To calculate the percentage activation, the average MFI of the SNAP-treated cells in each experiment was obtained. The MFI of each untreated sample was then divided by this and multiplied by 100 to obtain the percent activation.

**Fluorescence microscopy**. Imaging by fluorescence microscopy was performed using a ×60 1.3 NA silicone or ×40 0.9 NA air objective with a standard "Pinkel" DAPI/FITC/Cy3/Cy5 Filter set (Semrock). Emission was separated and captured using a 414/497/565/653 nm dichroic mirror and a quad-band bandpass emission filter between 503–515 and 614–804 nm. Images were analysed with Slidebook 6 (version 6.0.7 (25602), Intelligent Imaging Innovations, Denver, USA). For deconvolution, 10–16 slices were captured with a 0.33 μm step size, exported and deconvolved using the Richard-Lucy algorithm[48,49] with the CUDA[50] deconvolution plugin in ImageJ (version 1.52g, Wayne Rasband, National Institute of Health, USA). Pearson's correlation coefficient was calculated using the EzColocalization ImageJ plugin[51].

**Statistics and reproducibility**. Microscopy and in-gel fluorescence experiments were repeated at least twice with representative images shown.

**SNAP$_{Switch}$ chemical synthesis**. The quenched and attachable benzylguanine substrate was constructed via Fmoc solid-phase peptide synthesis on 2-chlorotrityl chloride resin (200 mg). Fmoc groups were removed using two treatments for 2 min and one treatment for 5 min with 20% piperidine in DMF. The resin was bubbled for 1 h in DCM with ethylenediamine (4 equiv., 50 mg, 0.83 mmol). $N(\alpha)$-Fmoc-$N(\varepsilon)$-azide-L-lysine (1.5 equiv., 0.11 g, 0.3 mmol) was attached with PyBOP (1.5 equiv., 0.16 g, 0.3 mmol) and DIPEA (2 equiv. to amino acid, 76 mg, 0.6 mmol) in DMF for 1 h. A PEG linker was then attached by adding Fmoc-N-amido-dPEG$_4$-acid (1.5 equiv., 0.15 g, 0.3 mmol) with PyBOP (1.5 equiv., 0.16 g, 0.3 mmol) and DIPEA (2 equiv., 76 mg, 0.6 mmol) for 30 min in DMF. After deprotection, Fmoc-BG-COOH (1.5 equiv., 0.17 g, 0.3 mmol) was attached with PyBOP (1.5 equiv., 0.16 g, 0.3 mmol) and DIPEA (2 equiv., 76 mg, 0.6 mmol) in DMF overnight before washing with DMF, DCM and then followed by deprotection.

Fifteen micrograms of resin was combined in a 1.6 mL Eppendorf tube with the quencher sQSY-21-COOH (0.4 equiv., 5 mg, 5.94 μmol), PyBOP (1.5 equiv.) and an excess of DIPEA (10 μL) in anhydrous DMSO for 23 h after rotating. The resin was transferred back to a solid-phase peptide synthesis column and washed once with DMSO, followed by washes with MeOH until the flow through was clear and then three times with DCM. The resin was dried under $N_2$ and the compound cleaved with 4 mL of 50% DCM and 50% HFIP with 10 μL TIPS for 2 h at room temperature. The mixture was drained into a round bottom flask using HFIP and the solvent evaporated under $N_2$. The compound was purified via HPLC with a water/MeCN 0.1% TFA gradient from 5 to 100% MeCN at a flow rate of 7 mL min$^{-1}$ over 45 min. The product mass was confirmed via LCMS.

To the entire fraction from HPLC (4.97 mg), Cy5-NHS (1 equiv.) was added with excess DIPEA (10 μL) in 200 μL DMSO and left overnight. The presence of the product mass was confirmed via LCMS and purified using 5–100% gradient over 45 min at a flow rate of 7 mL min$^{-1}$. The product mass was identified by LCMS, confirmed by HRMS and lyophilized. HRMS (ESI) $m/z$: calculated for $C_{108}H_{124}N_{18}O_{25}S_5$ [M + 3H]$^{+3}$ 745.2602, found 745.2635, calculated for [M + 2H]$^{+2}$ 1117.8882, found 1117.8887 and calculated for [M + 2Na]$^{+2}$ 1139.8701, found 1139.8704 (Supplementary Fig. 17). Purity was estimated by analytical HPLC as ~82% (Supplementary Fig. 18). SNAP$_{Switch}$ was reconstituted in DMSO to a concentration of 4 mM and stored at −20 °C.

**Fluorescence activation by the SNAP-tag in solution**. The effect of SNAP-tag excess on SNAP$_{Switch}$ was investigated using a fluorescence plate reader. SNAP$_{Switch}$ was initially diluted in PBS to 0.10 μM. Forty-five microlitres of this solution was then combined with 0 to 20 molar equivalents of SNAP-tag and PBS to bring the final volume to 65 μL in a 96-well clear bottom black polystyrene microplate. The fluorescence emission at 661 nm was obtained using a 646 nm excitation and was recorded every 10 min over 90 min total.

As SNAP-tag was at a much higher concentration than SNAP$_{Switch}$, pseudo first-order kinetics were assumed to apply. An initial fit of an exponential

one-phase association model (GraphPad Prism, Eq. (2)) to each concentration of SNAP-tag in Fig. 2c was used to obtain estimates for the plateau value. This was then used as a constraint for a further iteration (<2000 for this data set) and to obtain the observed rate constant, $k_{obs}$ (s$^{-1}$),

$$I = Y_0 + (\text{plateau} - Y_0)\left(1 - e^{(-k_{obs}t)}\right), \tag{1}$$

where $Y_0$ is the y-intercept and $t$ is time in seconds.

$k_{obs}$ was then plotted against SNAP-tag concentration and a simple linear regression was applied to obtain the second order rate constant, $k$ (M$^{-1}$ s$^{-1}$) (Supplementary Fig. 2),

$$k_{obs} = k[\text{SNAP-tag}]. \tag{2}$$

**Split-GFP assay**. The effect of the number of GFP$_{11}$ equivalents on fluorescence was investigated using a fluorescence plate reader. GFP$_{1-10}$ was initially diluted in PBS to 6 μM. Fifty microlitres of this solution was then combined with 0 to 0.33 molar equivalents of GFP$_{11}$ peptide (GL Biochem)[52] in 50 μL to bring the final volume to 100 μL in a 96-well clear bottom black polystyrene microplate. The fluorescence emission at 515 nm was obtained using a 470 nm excitation and the fluorescence was recorded every 5 min for the first hour, and then every 15 min after that.

**Fluorescence in-gel detection**. Samples were prepared by combining 5 μM SNAP-tag protein or PBS with 10 μM of SNAP$_{Switch}$ and 1 mM DTT in a final volume of 40 μL. The reaction was incubated at 37 °C for 30 min and then left to cool for 10 min before adding 6 μL of 0.2% 2-mercaptoethanol in sodium dodecyl sulfate-polyacrylamide gel electrophoresis loading dye[53]. Samples were then heated to 94 °C for 2 min and added to the wells of a pre-cast 12% polyacrylamide gel (Bio-Rad). The gel was run for 45 min at 120 V in running buffer (25 mM Tris, 250 mM glycine, 0.1% SDS at pH 8.3).

**Plasmid construction**. The empty transfection control plasmid pcDNA3.1(−) was purchased from Life Technologies. The TfR-SNAP fusion construct was modified from mEmerald-TFR-20 (a gift from Michael Davidson, Addgene plasmid #54278). Briefly, the sequence encoding SNAP-tag was PCR amplified from pSNAPf (New England BioLabs) with primers to append flanking BamHI and NotI restriction sites (forward: 5′-GGA TCC ACC GGT CGC CAC CAT GGA CAA AGA CTG CG-3′, reverse 5′-CGC GGC CGC TTA ACC CAG CCC AGG CTT GCC-3′). This PCR product was subsequently ligated into mEmerald-TFR-20 at the same sites, replacing the mEmerald coding sequence. A plasmid containing a mTurquoise2-H2A sequence (a gift from Dorus Gadella, Addgene plasmid #36207)[54] was PCR amplified to append BamHI and NotI restriction sites (forward: 5′-ACA GGA TCC ATG GTG AGC AAG GGC GAG GAG-3′ and reverse 5′-TGC GGC CGC GTT ATT TGC CTT T-3′). This PCR product was then ligated into pSNAPf.

The other lentiviral transfer plasmids were constructed as follows. Briefly, the SNAP-tag coding sequence from pSNAPf was subcloned into pCDH-EF1-MCS-IRES-Puro via NheI and BamHI to generate pCDH-EF1-SNAP-IRES-Puro (Cyto-SNAP). The SNAPf-mTurquoise2-H2A, pSNAPf-H2A and TfR-SNAP coding sequences were excised by restriction with NheI and NotI and subcloned into pCDH-EF1-MCS-IRES-Puro. All constructs were verified by DNA sequencing before use.

The sequence for transmembrane 106b (TMEM106b) was obtained from the Gene database of the National Center for Biotechnology Information[55] (Gene ID: 54664) and fused to the N or C terminus of the sequence for pSNAPf. The entire sequence was ordered as a plasmid from Twist Bioscience, inserted into pTwist Lenti SFFV Puro WPRE.

To generate plasmids for Escherichia coli expression, the SNAP-tag sequence was PCR amplified from pSNAPf with the following primers: forward, 5′-CTG TACTTCCAATCCAATGACAAAGACTGCGAAATGAAGCGCACCAC-3′ and reverse, 5′-CCGTTATCCACTTCCAATCCCTCGCAGACAGCGAATTAATTC CAGCA-3′, and were designed for overlap with pET His6 TEV LIC cloning vector (1B) (a gift from Scott Gradia, Addgene plasmid #29653). The pET His6 TEV LIC cloning vector (1B) was linearized at restriction site SspI and the SNAP-tag sequence was ligated into the plasmid using NEBuilder HiFi DNA Assembly Master Mix.

The sequence encoding GFP$_{1-10}$ for the split-GFP assay[52] was ordered as a gBlock from Integrated DNA Technologies and inserted into pET His6 MBP TEV LIC cloning vector (2M-T) (a gift from Scott Gradia, Addgene plasmid #29708) digested with NcoI and KpnI restriction enzymes and ligated with NEBuilder HiFi DNA Assembly Master Mix.

The following plasmids are available from Addgene: pCDH-EF1-TfR20-SNAP-IRES-Puro (Plasmid #138970), pCDH-EF1-SNAP-mTurquoise2-H2A-IRES-Puro (Plasmid #138968), pCDH-EF1-SNAP-H2A-IRES-Puro (Plasmid #138972), pCDH-EF1-SNAP-IRES-Puro (Plasmid #138969) and pET-HIS6-TEV-SNAP (Plasmid #138971)

**Stable cell line generation**. Stable cell lines (HEK-Cyto-SNAP, HEK-SNAP-H2A, HEK-TfR-SNAP, 3T3-Cyto-SNAP, 3T3-SNAP-H2A and 3T3-TfR-SNAP) were generated using lentiviral transduction. Lentivirus was generated in cells by

transfecting the transfer plasmid (pCDH series) together with packaging plasmids pMDLg/pRRE (Addgene #12251), pMD2.G (Addgene #12259) and pRSV-Rev (Addgene #12253). The viral supernatant was collected 48 h post-transfection and was filtered through a 0.45 µM filter. The target cells were seeded in a 24-well plate the day before, and 0.5 mL of filtered viral supernatant was applied.

At 48 h after the transduction, the media were replaced with culture media with 2 µg mL$^{-1}$ puromycin to perform the selection. After the transduced cells were fully grown, single-cell sorting was performed by incubating cells with 1 µM SNAP-Cell 647-SiR following the manufacturer's instructions. The sorting was done on the low–middle-intensity population. Homogeneous SNAP-tag expression within the generated cell line was confirmed by treating cells with 6.25 nM (H2A- and Cyto-SNAP) or 12.5 nM (TfR-SNAP) SNAP-Cell 647-SiR. The cells were incubated for 30 min, washed three times with DMEM with 10% FBS, and then incubated for a further 30 min in 400 µL media. All samples were then washed twice in DMEM with 10% FBS and detached using 200 µL TrypLE at 37 °C for 5 min. After detachment, 100 µL 1% bovine serum albumin in PBS was added to each well and the entire content transferred to a 96-well V-bottom plate. Cells were spun at 350 × g for 5 min and resuspended in PBS before analysis by flow cytometry (Supplementary Fig. 19).

**SNAP-tag and GFP$_{1-10}$ expression**. Proteins were expressed in *E. coli* and purified using immobilized nickel (SNAP-tag) or amylose (GFP$_{1-10}$) chromatography. Both were used without further modification. Protein concentration was estimated by measuring the UV–vis absorbance at 280 nm and using an extinction coefficients estimated using ProtParam[56].

**Estimation of SNAP$_{Switch}$ concentration**. The estimated extinction coefficient of the quenched SNAP-tag substrate was determined using the binary system[57] Beer–Lambert law:

$$A_n = \varepsilon_1 c_1 l + \varepsilon_2 c_2 l, \tag{3}$$

where $A_n$ is the absorbance at wavelength $n$, $l$ the path length (cm), $\varepsilon_m$ the extinction coefficient of species $m$ at wavelength $n$ (M$^{-1}$ cm$^{-1}$), $c_m$ the concentration of species $m$ at wavelength $n$ (M). Since both the fluorophore and quencher are conjugated to the same molecule, the concentration of each species is the same ($c_1 = c_2 = c$). Using a path length of 0.1 cm, Eq. (3) reduces to:

$$A_n = 0.1c(\varepsilon_1 + \varepsilon_2). \tag{4}$$

The extinction coefficient of each dye at the absorption maximum of the other dye was estimated by measuring the absorbance of a solution with known concentration of Cy5 at 661 nm and of sQSY-21 at 646 nm to give $\varepsilon_{QSY(646)}$ and $\varepsilon_{Cy5(661)}$. An estimate for the extinction coefficient of SNAP$_{Switch}$ was then obtained using Eq. (3) and the absorbance of the conjugate at either of these two wavelengths with the extinction coefficients at the absorption maximum of each dye ($\varepsilon_{Cy5(646)} = 271{,}000$ M$^{-1}$ cm$^{-1}$ and $\varepsilon_{QSY(661)} = 90{,}000$ M$^{-1}$ cm$^{-1}$).

**Protein labelling**. To label anti-CD44, anti-hTfR or anti-mTfR, 50 µg of protein was diluted in 250 µL of PBS and incubated with 10 equiv. of 1 mg mL$^{-1}$ DIBO-NHS, AF488-NHS or 2 mg mL$^{-1}$ BDP-FL and Cy5-NHS dissolved in DMSO for 2 h at 4 °C. Unreacted DIBO or dye was removed using a 0.5 mL Zeba Spin Desalting Column, 7 K molecular weight cut-off, pre-equilibrated with PBS, according to the manufacturer's instructions. For SNAP$_{Switch}$ labelling, 1.25 equiv. of SNAP$_{Switch}$ was added and incubated overnight at 4 °C before excess was removed using additional Zeba Spin Desalting Columns.

The degree of labelling (DOL) was estimated by dividing the concentration of SNAP$_{Switch}$ approximated using Eq. 3 by the protein concentration as determined using the absorbance at 280 nm with the Beer–Lambert law and an extinction coefficient for the protein ($\varepsilon_{antibody} = 210{,}000$ M$^{-1}$ cm$^{-1}$). The degree of AF488, BDP-FL or Cy5 labelling was calculated using the Beer–Lambert law, the absorbance at 495, 503 or 646 nm and an extinction coefficient of $\varepsilon_{AF488} = 73{,}000$ M$^{-1}$ cm$^{-1}$, $\varepsilon_{BDP-FL} = 80{,}000$ M$^{-1}$ cm$^{-1}$ or $\varepsilon_{Cy5} = 271{,}000$ M$^{-1}$ cm$^{-1}$, respectively.

**Oligonucleotide labelling**. The custom 20-mer oligonucleotide modified with dibenzocyclooctyne (DBCO) (sequence: 5′-TCA GTT CAG GAC CCT CGG CTDBCO-3′) was purchased from IBA Life Sciences, reconstituted in nuclease free water at a concentration of 600 µM and stored at −20 °C. The amine-modified version (sequence: 5′-TCA GTT CAG GAC CCT CGG CT-amino modifier-3′) was purchased from Integrated DNA Technologies and reconstituted using the same conditions. For DBCO-modified oligonucleotide labelling, $1.2 \times 10^{-8}$ mol of the sequence was incubated with SNAP$_{Switch}$ (2 equiv., 24 nmol) and AF488-Azide (1.25 equiv., 15 nmol) overnight at 4 °C. Unconjugated dye was spun through 7 K molecular weight cut-off Zeba Spin columns until free dye no longer permeated throughout the entire resin.

For amine-modified oligonucleotide labelling, $1.2 \times 10^{-8}$ mol of the sequence was incubated with DIBO-NHS (5 equiv., 60 nmol) for 2 h at 4 °C. Excess DIBO was removed using an Amicon Ultra 0.5 mL Centrifugal Filter with a 3 K molecular weight cut-off. Three 5-min washes were performed using of 500 µL PBS and spinning the units at 13,000 × g. SNAP$_{Switch}$ (0.25 equiv.) and AF488-Azide (0.25 equiv.) were then added and incubated overnight at 4 °C. Unreacted dye was then

removed using ethanol precipitation where 1 mL of 9:1 v/v ethanol to 5 M sodium acetate solution was added before spinning the sample for 20 min at 4 °C. The supernatant was discarded and 500 µL of −20 °C 70% ethanol was added to the sample without disturbing the pellet. The sample was centrifuged for 20 min at 4 °C, the supernatant was discarded, and the sample was air-dried. The absorbance of the conjugate at 260 nm was measured and DOL was calculated using the previously described extinction coefficients and $\varepsilon_{DNA} = 181{,}000$ M$^{-1}$ cm$^{-1}$ for the oligonucleotide.

**Anti-transferrin receptor antibody association**. Wild-type 3T3 cells or cells stably expressing TfR-SNAP, Cyto-SNAP, TMEM106b-SNAP or SNAP-TMEM106b were seeded in 24-well plates 1 day prior at 100,000 cells per well in 400 µL DMEM supplemented with 10% FBS and penicillin/streptomycin. Proteins were incubated with the cells at 5 µg mL$^{-1}$ for 1–4 h before washing twice with PBS and detached with 100 µL TrypLE at room temperature. After 5 min, 200 µL 1% bovine serum albumin in PBS was added and the entire volume was transferred to 96-well V-bottom plates. Cells were pelleted at 350 × g for 5 min and resuspended in PBS for analysis by flow cytometry.

**Pre-activated SNAP$_{Switch}$ conjugates**. Wild-type 3T3 cells or cells stably expressing TfR-SNAP were seeded in 24-well plates 1 day prior at 100,000 cells per well in 400 µL DMEM supplemented with 10% FBS and penicillin/streptomycin. Anti-mTfR dual labelled with AF488 and SNAP$_{Switch}$ (6.2 µg) was incubated in 1 mM β-mercaptoethanol in PBS (final volume = 93 µL) with or without 30 equiv. of SNAP-tag to the SNAP$_{Switch}$ DOL for 1 h at 37 °C. Anti-mTfR was then added to cells at 2.5 µg mL$^{-1}$ and incubated for 1 h at 37 °C and 5% CO$_2$. The cells were washed twice in PBS and detached with 200 µL TrypLE for 5 min at room temperature before adding 100 µL 1% bovine serum albumin in PBS. The cells were transferred to a 96-well V-bottom plate and spun at 350 × g for minutes. The supernatant was removed, and the cells were resuspended in 100 µL before analysis by flow cytometry.

**Lipofectamine 3000 induced endosomal escape**. HEK cells were seeded in 24-well plates 1 day prior at 50,000 cells per well in 400 µL DMEM supplemented with 10% FBS and penicillin/streptomycin. For H2A-mTurquoise-SNAP experiments, cells were transfected with 500 ng DNA or an empty plasmid (pCDNA31(−)) using Lipofectamine 3000, following the manufacturer's protocol. A measure of 0.75 µL of Lipofectamine 3000 reagent and 1 µL of P300 reagent was used per well. After 16 h, cells were washed three times in DMEM + 10% FBS before further use. For experiments with TfR-SNAP, cyto-SNAP and H2A-SNAP stable cell lines were used.

Cells were transfected with complexes containing either unlabelled oligonucleotide or oligonucleotide labelled with SNAP$_{Switch}$ and Alexa Fluor 488. Samples consisted of 500 ng DNA total with 250 ng each of unlabelled and labelled oligonucleotide (also containing an unlabelled fraction) and were transfected using the same amounts of Lipofectamine 3000 and P3000 previously described. The transfection was performed at five time points between 0.5 and 16 h. For SNAP-tag labelling, the media were replaced with 250 µL DMEM with 10% FBS containing 0.6 µM SNAP-Cell SiR 647 (diluted from 0.6 mM in DMSO). The cells were incubated for 30 min, washed three times with DMEM with 10% FBS, and then incubated for a further 30 min in 400 µL media. All samples were then washed twice in DMEM with 10% FBS and detached using 200 µL TrypLE at 37 °C for 5 min. After detachment, 100 µL 1% bovine serum albumin in PBS was added to each well and the entire content was transferred to a 96-well V-bottom plate. Cells were spun at 350 × g for 5 min and resuspended in PBS before analysis by flow cytometry.

To test cell viability, cells transfected with 500 ng empty plasmid for 4, 8, and 16 h or cells without transfection were washed twice in PBS and incubated with or without a 1:250 dilution of Zombie Yellow Viability dye in 200 µL PBS for 15 min at room temperature, and prepared according to the manufacturer's protocol. Cells were then detached and analysed as above. As a positive control, 6 wells were combined, spun at 350 × g for 5 min, resuspended in 600 µL PBS and split into two 1.6 mL tubes. One tube was incubated at 65 °C for 5 min to induce apoptosis, while the other remained on ice. The viability dye was then added at a 1:250 dilution for 15 min at room temperature. The cells were then spun, and finally resuspended in 1% BSA/PBS before washing an additional two times with PBS before analysis by flow cytometry. The average MFI of cells without Zombie treatment was subtracted from the MFI of samples with the dye.

Cells transfected with H2A-mTurquoise-SNAP and positive for expression were used to compensate for spillover into the AF488 channel. Spillover of mTurquoise and AF488 fluorescence into the Cy5 channel was negligible.

**Antibody colocalization and TMEM106b imaging**. 3T3 cells stably expressing TfR-SNAP were seeded 1 day prior at 10,000 cells per well in 100 µL in 96-well plates. Fluorescently labelled anti-hTfR, anti-mTfR or anti-CD44 was then added at 5 µg mL$^{-1}$ and incubated for 1–4 h at 37 °C. For labelling of SNAP-tag, the media were replaced with 100 µL DMEM + 20% FBS containing 5 µM SNAP-Cell 505-Star (or 2 µM SNAP Surface Alexa Fluor 647) and incubated for 1 h at 37 °C and 5% CO$_2$. The cells were then washed in media three times. SNAP-cell-treated samples were then incubated for a further 30 min in the media. All samples were

then washed twice in PBS or three times in FluoroBrite + 10% (for SNAP-tag imagining experiments) and the media were replaced with FluoroBrite + 10% FBS with or without Hoechst for 10 min prior to imaging.

**Lipofectamine 3000 endosomal escape imaging**. HEK293 cells were seeded at 20,000 cells per well in an 8-well chamber slide 1 day prior in 400 µL DMEM supplemented with 10% FBS and 10,000 penicillin/streptomycin. Cells were transfected with 500 ng Cyto-SNAP, H2A-SNAP or an empty plasmid with Lipofectamine 3000 by following the manufacturer's protocol. A measure of 0.75 µL of Lipofectamine 3000 reagent and 1 µL of P300 reagent was used per well.

After 16 h, cells were washed three times in DMEM + 10% FBS and transfected again with 3100 ng oligonucleotide labelled with SNAP$_{Switch}$ (also containing an unlabeled fraction) and were transfected using the same amounts of Lipofectamine 3000 and P3000 reagent as the previous transfection. The cells were incubated for an additional 16 h before washing the cells twice with cold FluoroBrite with 10% FBS and then left for 10 min on ice. The membrane was then stained with wheat germ agglutinin-AF488 at a final concentration of 1 µg mL$^{-1}$ for 5 min before washing an additional two times in FluoroBrite with 10% FBS.

To compare the location of SNAP$_{Switch}$ signal to SNAP-tag labelled with fluorescent SNAP substrates, cells transiently expressing Cyto-SNAP or H2A-mTurquoise-SNAP were treated with 50 µL of 5 µM SNAP-Cell TMR-Star or 3 µM SNAP-Cell SiR 647 in culture media for 30 min at 37 °C, 5% $CO_2$ before washing the cells three times in DMEM with 10% FBS. The media were replaced, and the cells were incubated for a further 30 min before replacing the media with FluoroBrite with 10% FBS before imaging.

**Reporting summary**. Further information on research design is available in the Nature Research Reporting Summary linked to this article.

## Data availability

The source data underlying all the figures of this study are available as Source Data files. The raw data is available from the corresponding author upon request. Source data are provided with this paper.

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

## Acknowledgements

We thank Dr. Jason Dang (Monash University, Melbourne) for his assistance with HPLC, LCMS and HRMS.

## Author contributions

L.I.F., A.P.R.J. and B.G. designed the study. L.I.F, L.A and B.G. performed the chemical synthesis. M.C., D.Y. and J.J.R. performed the cloning. J.J.R. performed the GFP experiments. L.I.F. and A.P.R.J. analysed the data and wrote the manuscript.

## Competing interests

The authors declare no competing interests.
