## [Peer Review File · Nature Communications]

Reviewers' comments:

Reviewer #1 (Remarks to the Author):

The authors described a quencher-releasing mediated fluorogenic SNAP substrate, named SNAP-switch, to label biomolecules like antibody or oligonucleotide and track their location or trafficking in cells. However, similar works based on SNAP-tag technology had been reported a lot, since Kai Johnsson reported SNAP-tag in 2003. For example, the interaction between fluorescent substrate and SNAP-labeled target protein or antibody is traced by fluorescence imaging, including the development of new SNAP fluorescence probes for super-resolution spatial resolution and time-resolved. SNAP-tag technology consists of two parts, SNAP-labeled target protein and SNAP fluorescent substrate. Because the efficiency of SNAP on protein transfer is affected by different conditions, most researchers use SNAP technique to study the location of the target protein and the interaction between the fluorescent labeled protein and other molecules. The focus on SNAP fluorescent substrate is to consider the fluorescence change between substrate and SNAP before and after the reaction, so as to indicate the position of substrate molecule in cells and the interaction with SNAP. However, the premise of such research is that the fluorescent substrate has the fluorescence signal which is not affected by the environment and the stable emission to facilitate the time-resolved tracking imaging, as well as the fast covalent connection reaction between the substrate and SNAP to ensure the real-time tracking of time resolution. However, the reaction rate of SNAP-switch molecule with SNAP is very slow, and it needs to react for 90 mins with 20 equivalents of SNAP. Also, the photostability of cy5 cannot guarantee the dynamic tracking in cells.

In addition, there are other technical problems limiting this method to track the trafficking the cells.

1. SNAP is located in cells, while the distribution of fluorescent substrates in the cell does not exist in many places where SNAP does not exist. Therefore, the fluorogenic signal can only indicate the presence of the substrate closed SNAP, but not whether the substrate exists in other parts of the cell.

- 2, The reaction of substrate with SNAP is highly structurally selective. SNAP is fused to antibody, and fluorescent dye is conjugated to antigen. The production of fluorescence indicates the interaction between SNAP and substrate, but it does not indicate that the recognition of antibody-antigen is the first. Indeed, after the interaction between antibody and antigen, it is likely that the space distance between SNAP and substrate is too far to react each other. Therefore, I recommend to reject this paper.

Other questions:

1. The SNAP-switch structure should be shown in figure 1. The principle shown in figure 1 is too complex to understand.

2. The fluorescence spectra and kinetics of the reaction between SNAP-switch and SNAP in buffer solution need to be proved. The fluorescence changes in gel experiments in figure 2 is not sufficient. 3. The Pearson correlation coefficient needs to be given in fluorescence imaging.

Reviewer #2 (Remarks to the Author):

This is interesting and potentially very useful research with applications across the biological and physical sciences. The design of the SNAPswitch is carefully tested and explained, and I can see many uses for such a technique, but I feel that the illustrative application in analysing transfection (certainly a highly complex process in desperate need of such novel probes) is not adequate to depict the strengths of the reported technique. On balance, there is no doubt the labelling/activation approach can be useful and the data presented up to and including Figure 4 clearly support the success and potential for the technology. I would therefore recommend the article is suitable for a more specialised journal and not for Nature Comm in this particular instance.

The data in Figure 5 is not convincing. While some identification of localised pDNA is depicted, evidence of transfection itself is limited and may be confounded by toxic effects of membrane disruption, cell division, and the multitude of barriers involved. It is perhaps unsurprising that pDNA has been identified in both cytoplasm and nucleus, considering that the cell cycle has not been arrested – this potentially allows 'backwards' pDNA diffusion during mitosis and is not necessarily evidence for endosomal escape. It is unclear whether the cell depicted in Figure 5d (again relating to the claim of endosomal escape) is destined for successful transfection or apoptosis; there is a fine balance in transfection between efficiency and toxicity, which may manifest in specific cells while there is little change in viability of the overall population. Coupled with the fact that identifying pDNA in the nucleus is not necessarily the final determinant of transfection [e.g., J Control Release 135, 166 (2009)], it seems to me that transfection raises more questions than it answers and would need to be more thoroughly investigated.

I suggest some further experiments in line with the transfection ambiguities, as the conclusion that pDNA is lost at each stage of the transfection process is not new and undersells what is otherwise an exciting new technology.

Suggested further experiments:

The claim that pDNA dissociates from Lipofectamine prior to nuclear translocation is not supported by any experimental data except activation by the SNAPswitch. I imagine the SNAPswitch could still be activated if the lipoplex disassembled in the nucleus, and my understanding is that dissociation of lipoplexes prior to nuclear entry has not yet been conclusively demonstrated. Can you show that there is no SNAP activation for pDNA complexes protected by Lipofectamine in solution and/or at early stages of internalisation as you did so carefully in e.g., Figure 3? What if there is some uncomplexed DNA bound to the vesicle surface, or partially encapsulated DNA? Perhaps various transfection reagents could be tested at different concentrations to confirm that the escape phenomenon in Figure 5d is widespread, correlates with transfection efficiency, and not with cell death.

Can the use of a circular plasmid encoding a reporter gene rather than a noncoding oligo be used to correlate between cyto-SNAP and H2A-SNAP positive plasmids and subsequent success/failure in transfection?

Other major comments:

There is gathering evidence that Lipofectamine 2000 can evade microtubule-dependent trafficking and therefore the endosomal pathway [Sci Rep 6, 25879 (2016)] and so the claim of endosomal escape may require some qualification.

Was the Lipofectamine to DNA ratio optimised as directed in the protocol? Flow data will help greatly here to confirm high transfection with minimal cell death.

The opening sentence in para. 3 is potentially confusing. I think perhaps the authors mean fluorescent fusions have traditionally been used to detect and track proteins of interest rather than label subcellular structures.

HPLC methods: 0.1% TFA is written twice.

More flow cytometry data is required in the supplementary information to illustrate gating and representative results for all experiments.

The clause citing Reference 3 does not refer to transport of inbound biomolecules as does the rest of the sentence.

Please clarify which Lipofectamine reagent is used in the final paragraph of the introduction section and in para. 2 of the Discussion.

Sensor characterization in solution section: You refer to a 'significant difference' but no statistics are reported. Perhaps 'observable difference' would be less ambiguous here?

Discussion, para. 2: You refer to TfR everywhere except here, where CD71 is used. Perhaps this could be more consistent.

Figure captions concerning 'normalisation' – the images should have been collected with the same exposure settings and been processed identically making this statement unnecessary. What exactly are the data normalised to?

Supp figures 6, 7: Are the images labelled as DIC really DIC? If so, the bias should have been increased to obtain better images as contrast is very low making it almost impossible to distinguish cells from the background.

Figure 5: What is the timepoint at which the images were taken?

Typographical/grammatical corrections:

Abstract: SNAPswitch

Discussion, para. 1: Missing full stop; 'processes such occurs in'

'The spatial overlap ... is measured by microscopy and statistical analysis (...) is then used'

Supplementary Figure 8 is missing labels (a), (b) and panels seem to be in a different order than the caption suggests.

Reviewer #3 (Remarks to the Author):

FitzGerald et al., have developed a localization sensor (SNAPSwitch) that is activated when it comes into close proximity to a SNAP-tagged protein. SNAPSwitch enables quantitative detection of protein, nucleic acid and nanoparticle trafficking to locations of interest within live cells. With this approach authors followed the trafficking of DNA nanoparticle complexes travelling from endosomes into the cytosol and into the nucleus and antibody targeted to the transferrin (CD71) or hyaluronan (CD44) receptor is initially sorted into different compartments following endocytosis. Even though it is a topical area of review, the present tags have much more better resolution and the current study did not show any data to suggest use of this technology for high-throughput analysis. The manuscript in the current form is unsuitable for publication and with utmost respect for the work presented have to unfortunately reject it.

1. My first issue is that the trafficking of transferrin or transferrin antibody follows a very quick uptake within 5 mins with recycling within 15 mins. The quality of images just does not allow to visualize punctate with high resolution which can be even seen with usual tag. For example, when transferrin internalizes a vesicular structure is observed and within 10 mins a tubulovesicular structure appears indicating recycling endosomes and large vesicular lysosomes can be visualized in 20 mins. Generally, the experiments are pulse chase, starting the incubation at 4 degrees with warming the media and performing imaging. Again, I was not able to visualize any structural minutiae with this tech which can be seen even with just labeled transferrin. The current method does not show any superiority in imaging.

2. The similar issue is with experiments done for endosomal escape. First lipofectamine 3000 based lipoplexes do not form stable nanoparticles and due to very high polydispersity of lipofectamine a lot of aggregation is generally observed on plates. Having said that there have been lot of studies showing trafficking of these complexes and again the image quality was not high enough to pinpoint any differences. The time scales chosen for the events are generally associated with microscopy techniques that generally have low resolution. The state of the art tools have quantified the escape (for siRNA) that has been relatively low and DNA amount is even lower which again was very hard for me to make any visual judgment. There is a cell membrane label, associated with AF488, some cytosolic amounts and little if any label that may not still be attached with the nanoparticle. Again H2A snap images are confusing (especially the membrane localization) So, based on previous publication and quality of images and number of cells imaged to quantify diminishes enthusiasm for the current manuscript

3. Finally, my biggest critique is that authors are messaging the work to be useful for highthroughput analysis without providing any data. Furthermore, tools like CLIP or other tools with peptide tags (Proc Natl Acad Sci U S A. 2018 Dec 18; 115(51):12961-12966. doi:

10.1073/pnas.1808626115. Epub 2018 Dec 5) and FRET based quenchers have been shown to give high resolution images (Integr Biol (Camb). 2013 Jan;5(1):224-30. doi: 10.1039/c2ib20155k.) and many others have been shown to be used prior. The idea presented here could have been exciting if it could decipher anything new in terms of trafficking or gave a better quality image that can be interpreted easily.

We thank the reviewers for their comments. We have addressed all the points raised as outlined below.

Reviewer 1:

1.1 However, similar works based on SNAP-tag technology had been reported a lot, since Kai Johnsson reported SNAP-tag in 2003. For example, the interaction between fluorescent substrate and SNAP-labeled target protein or antibody is traced by fluorescence imaging, including the development of new SNAP fluorescence probes for super-resolution spatial resolution and time-resolved.

We think the reviewer may not have fully understood the concept of our localization assay. All the published SNAP-tag assays rely on co-localization analysis. Co-localization analysis is inherently low throughput, as a high-resolution microscope can only acquire data for 5-20 cells per image and requires time consuming co-localization analysis. It is very rare for more than 100 cells to be analysed, and typically less than 10 cells are analysed. In contrast, our localization assay uses flow cytometry, which can acquire >10,000 cells per second. All the data in this paper is based on the analysis of >50,000 cells. To make this point clearer we have added the following sentence to the introduction.

“In addition, traditional colocalization analysis is low-throughput, as the number of cells that can be analyzed is limited (typically 10 – 100s of cells over hours) compared to other techniques such as flow cytometry (thousands of cells per second).”

Live cell super-resolution techniques (STED, SIM) are limited to a resolution of ~100 nm. Fixed cell super-resolution imaging techniques (STORM/PALM) have a resolution of >20 nm but have much longer acquisition times (up to 10 min per image). However, even this resolution would not be enough to determine which side of a membrane a material is located. To demonstrate the spatial resolution that can be resolved with our localization assay, we have performed additional experiments and included additional section “Resolving Membrane Orientation of Cargo” and an additional figure (Figure 5), that shows the SNAP_{Switch} can determine if material is on the luminal or cytosolic side of an endosomal membrane.

1.2 SNAP-tag technology consists of two parts, SNAP-labeled target protein and SNAP fluorescent substrate. Because the efficiency of SNAP on protein transfer is affected by different conditions, most researchers use SNAP technique to study the location of the target protein and the interaction between the fluorescent labeled protein and other molecules. The focus on SNAP fluorescent substrate is to consider the fluorescence change between substrate and SNAP before and after the reaction, so as to indicate the position of substrate molecule in cells and the interaction with SNAP. However, the premise of such research is that the fluorescent substrate has the fluorescence signal which is not affected by the environment and the stable emission to facilitate the time-resolved tracking imaging,

The design of the SNAP_{Switch} overcomes the limitations of sample environment raised by the reviewer. When the SNAP_{Switch} is turned on, the fluorescent dye remains attached to the material of interest (e.g. protein, DNA or nanoparticle). i.e. Once activated, the material fluoresces like it is labelled with a convention fluorescent dye.

Unlike fluorescent proteins such as GFP, Cy5 is a pH insensitive fluorophore, so different acidic environments (e.g. the lysosome) will not change the fluorescence of the sensor.

Furthermore, the SNAP_{Switch} is a ‘kiss and run’ system, where the labelled material is free to be trafficked in the cell after it has been switched on.

We have revised the manuscript to include the following paragraph to make these points clearer.

“After activation, the fluorescence is permanently switched on and the sensor remains attached to the material of interest. These features of SNAP_{Switch} offer significant advantages over typical image-based colocalization analysis and other assays for endosomal escape such as split-GFP.²⁵ Signal from SNAP_{Switch} accumulates over time as more interactions occur enabling

quantification of material transitioning through specific locations of the cell, such as cargo passing through an endosome into the cytosol. The fluorophore is pH-insensitive within a range relevant to the endocytic pathway (pH 4-10)²⁶ while most GFP-variants are not,²⁷ which is an issue where endocytic processes are involved due to acidification endosomes. Finally, the labelled material is not anchored to the SNAP-tag after an interaction which could potentially block further trafficking. This means subsequent trafficking of the material can be observed either in the same sample or at different time points, using flow cytometry to avoid photobleaching.”

1.3 ... as well as the fast-covalent connection reaction between the substrate and SNAP to ensure the real-time tracking of time resolution. However, the reaction rate of SNAP-switch molecule with SNAP is very slow, and it needs to react for 90 mins with 20 equivalents of SNAP.

The reviewer raises an important point about the reaction rate and molecular excess of SNAP-tag. The ex-vivo solution measurements are performed at a relatively high dilution, due to the sample volume required for the plate reader. These concentrations are much lower than the concentrations that occur inside a cell. At cellular concentrations, SNAP_{Switch} is switched on much faster. To demonstrate this, we have included additional data on the rate constant of this reaction (Figure 2c, SI Figure 2). We determined that the rate constant for SNAP_{Switch} turn on is an order of magnitude faster than click chemistry. We have revised the manuscript as follows to address this point.

“Activation of the sensor was rapid, with the second order rate constant was estimated to be $111 \pm 5 \text{ M}^{-1} \text{ s}^{-1}$ (SI Figure 2). This is significantly faster than most bio-orthogonal click reactions (e.g. strain-promoted alkyne-azide cycloadditions (SPAAC)), which typically range from 0.1 - 10 $\text{M}^{-1} \text{ s}^{-1}$,²⁹ and is similar to the activation of non-attachable quenched SNAP-tag substrates.^{24,28”}

We have also performed additional experiments to compare the activation of SNAP_{Switch} with a split-GFP reporter system and shown that SNAP_{Switch} a) activates faster and b) requires less equivalents of the activating component (SI Figure 3). The split-GFP system has been used extensively in literature and for detection of endosomal escape. The results here show that our localization assay is 1) more sensitive, 2) accumulates signal over time allowing for fleeting interactions to be detected and, 3) does not prevent subsequent cellular trafficking.

We have revised the manuscript to include the follow paragraph to address this point.

“To demonstrate SNAP_{Switch} has improved sensitivity compared to other fluorescence assays, we investigated the signal generated from a split-GFP complementation system, which has previously been used to detect cytosolic delivery of cell penetrating peptides (CPPs).^{25,30,31} Here, GFP is split into a large (GFP₁₋₁₀) and small (GFP₁₁) fragment, rendering it almost non-fluorescent until an interaction occurs between the two fragments. With 10 equivalents of protein (SNAP-tag or GFP₁₋₁₀) to its interaction partner (SNAP_{Switch} or GFP₁₁), a significant change above background was seen within 10 minutes ($p < 0.05$, unpaired t-test) for SNAP_{Switch} (Figure 2c), while split-GFP took 60 minutes for a significant signal to be detected ($p < 0.05$, unpaired t-test) (SI Figure 3). After 90-minutes, the SNAP_{Switch} signal was 7-fold higher than the control while split-GFP increases was only 1.5-fold higher.”

1.4 Also, the photostability of cy5 cannot guarantee the dynamic tracking in cells.

A significant advantage of our localization assay is that we do not need to image the same cell over an extend period of time to determine the cellular trafficking. This overcomes the limitation of photobleaching, which is a limitation of all fluorescent dye colocalization studies. The SNAP_{Switch} signal accumulates over time, which means we can sample a population of cells at different time points using flow cytometry.

We have revised the manuscript as outlined in response to point 1.2 above to explain this concept more clearly.

1.5 In addition, there are other technical problems limiting this method to track the trafficking the cells.

1. SNAP is located in cells, while the distribution of fluorescent substrates in the cell does not exist in many places where SNAP does not exist. Therefore, the fluorogenic signal can only indicate the presence of the substrate closed SNAP, but not whether the substrate exists in other parts of the cell.

Again, we think the reviewer may not have fully understood our localization assay. Unlike other SNAP imaging techniques, we are not seeking to determine where all the SNAP-tagged proteins are. We are determining where the SNAP_{Switch} labelled material goes. This is an important distinction. For example, if we wish to detect endosomal escape, we do not want to measure how much cytosolic SNAP-tag protein there is. We want to measure how much SNAP_{Switch} labelled material becomes fluorescent.

We are also able to track where material of interest is before the SNAP_{Switch} is turned on by dual labelling the material with a second fluorophore. An example of this approach is seen Figure 6b, c where association with cells can be seen from the Alexa Fluor 488 signal before SNAP_{Switch} turn on is observed in various subcellular locations.

1.6 2, The reaction of substrate with SNAP is highly structurally selective. SNAP is fused to antibody, and fluorescent dye is conjugated to antigen. The production of fluorescence indicates the interaction between SNAP and substrate, but it does not indicate that the recognition of antibody-antigen is the first. Indeed, after the interaction between antibody and antigen, it is likely that the space distance between SNAP and substrate is too far to react each other. Therefore, I recommend to reject this paper.

The reviewer is correct that SNAP-tag is highly selective, but they are incorrect to say we are using the localization assay to directly detect receptor binding. At no stage in this paper did we fuse SNAP-tag to an antibody. SNAP-tag was fused to either TfR, H2A, TMEM106b or it was expressed in the cytosol.

The SNAP_{Switch} does not switch on due to receptor binding. It switches on when SNAP_{Switch} comes into close proximity to a SNAP-tagged protein. For TMEM106b-SNAP, CytoSNAP and NuclearSNAP, no ligand receptor interaction occurs, SNAP_{Switch} switches on simply due to the presence of SNAP-tag in the local environment. We have added the following sentences to clarify this point.

“Activation of SNAP_{Switch} is not a direct result of SNAP_{Switch} labelled antibody binding to a SNAP-tagged receptor but occurs when SNAP_{Switch} labelled antibody comes into close proximity with a SNAP-tagged receptor. The SNAP-tagged receptor that activates the SNAP_{Switch} will likely not be the receptor that the antibody binds to, as the distance between the SNAP_{Switch} and SNAP-tag will be too great.”

1.7 The SNAP-switch structure should be shown in figure 1. The principle shown in figure 1 is too complex to understand.

We have simplified the figure 1 to make the concept easier to understand and include the structure of the SNAP_{Switch}.

1.8 The fluorescence spectra and kinetics of the reaction between SNAP-switch and SNAP in buffer solution need to be proved. The fluorescence changes in gel experiments in figure 2 is not sufficient. 3. The Pearson correlation coefficient needs to be given in fluorescence imaging.

The absorbance and fluorescence spectra of the SNAP_{Switch} have been included (SI Figure 2). We have also provided the rate constant for the solution experiment (SI Figure 3).

We have included the Pearson correlation coefficient for the fluorescence images.

Reviewer 2:

We thank the reviewer for highlighting the localization assay is “exciting new technology”, “with applications across the biological and physical sciences” and will have “many uses”.

2.1 *The data in Figure 5 is not convincing. While some identification of localised pDNA is depicted, evidence of transfection itself is limited and may be confounded by toxic effects of membrane disruption, cell division, and the multitude of barriers involved.*

Figure 5 (now Figure 6 in the revised manuscript) was not intended to show transfection. Figure 6 intends to show the trafficking of DNA cargo that has been delivered to cells by lipofectamine. The “toxic effects of membrane disruption, cell division, and the multitude of barriers involved” caused by lipofectamine cannot be decoupled from the use of lipofectamine as a delivery system, however we have performed additional experiments to show the cells are viable (SI Figure 10).

We have revised the manuscript as follows to clarify this point.

“The increase in signal is not due to cytotoxic effects of Lipofectamine, as cells transfected at different time points and treated with a viability stain had similar viability (MFI ~ 11) to non-transfected cells (MFI ~8.4), while the signal from apoptotic cells was approximately 100-fold higher (MFI ~ 1000) (SI Figure 10).”

2.2 *It is perhaps unsurprising that pDNA has been identified in both cytoplasm and nucleus, considering that the cell cycle has not been arrested – this potentially allows ‘backwards’ pDNA diffusion during mitosis and is not necessarily evidence for endosomal escape.*

The reviewer is correct to suggest that ‘backwards’ diffusion during mitosis could occur. We thank the reviewer for highlighting this point and have revised the manuscript as follows.

"It is possible that the DNA cargo comes in contact with the SNAP-tagged H2A during mitosis (when the nuclear membrane brakes down) and that direct delivery of DNA through the nuclear membrane does not occur.²⁶ Regardless of the mechanism, SNAP_{Switch} is able to determine if and when the delivered DNA can access the nuclear components of the cell."

2.3 *It is unclear whether the cell depicted in Figure 5d (again relating to the claim of endosomal escape) is destined for successful transfection or apoptosis; there is a fine balance in transfection between efficiency and toxicity, which may manifest in specific cells while there is little change in viability of the overall population. Coupled with the fact that identifying pDNA in the nucleus is not necessarily the final determinant of transfection [e.g., J Control Release 135, 166 (2009)], it seems to me that transfection raises more questions than it answers and would need to be more thoroughly investigated.*

While the reviewer is correct that it is not possible to know if the cells depicted in 6d-f are destined for transfection or apoptosis, this is the case with any microscopy data. However, the advantage of our localization assay is that we can use flow cytometry to track localization over time, analysing thousands of cells. Because the flow cytometry data in Figure 6g consists of the analysis of >50,000 cells, this limits the chance of our data being skewed by analysing cells that a destined for apoptosis, rather than transfection. The additional data we have included in SI Figure 10 shows that the cells are not apoptotic and are viable.

We have revised the manuscript as follows to clarify this point.

“As with most microscopy data used to determine localization, it is difficult to know if these cells demonstrating escape were fated for transfection or apoptosis. However, the compatibility of SNAP_{Switch} with flow cytometry allows for the analysis of thousands of cells, limiting the chance of data being skewed by analyzing cells that would undergo apoptosis.”

The reference the reviewer refers to says that as the number of plasmids per nucleus increases to ~3,000, the transfection efficiency increases. Beyond this value, any further increase is marginal. While the reviewer is correct that the amount of plasmid in the nucleus may not be the final

determinant of transfection efficiency, delivery of plasmid into the nucleus is still required for expression. When solely relying on end point assays, such as gene expression, if gene expression doesn't occur, then it isn't clear which step in the transfection pathway has failed or is inefficient. With the SNAP_{Switch} we can pin point which step in the trafficking pathway has failed. To make this clearer we have added the following sentence:

"Unlike endpoint assays, which give a yes or no answer for if delivery has occurred, the SNAP_{Switch} assay can determine the point in the trafficking pathway that has failed or is inefficient."

2.4 Suggested further experiments:

The claim that pDNA dissociates from Lipofectamine prior to nuclear translocation is not supported by any experimental data except activation by the SNAPswitch. I imagine the SNAPswitch could still be activated if the lipoplex disassembled in the nucleus, and my understanding is that dissociation of lipoplexes prior to nuclear entry has not yet been conclusively demonstrated.

As highlighted in response to point 2.2 above, the reviewer is correct that activation of SNAP_{Switch} by NuclearSNAP could occur during mitosis. We have reworded the manuscript to reflect this, as outlined in response to point 2.2 above. We also note, that while contact with H2A could have occurred while the DNA was complexed with lipofectamine, Figure 5e (6e in revised manuscript) does not show punctate fluorescence, suggesting that the DNA is no longer complexed with lipofectamine.

2.5 Can you show that there is no SNAP activation for pDNA complexes protected by Lipofectamine in solution and/or at early stages of internalisation as you did so carefully in e.g., Figure 3? What if there is some uncomplexed DNA bound to the vesicle surface, or partially encapsulated DNA?

We believe there will be activation of SNAP_{Switch} by SNAP-tag while complexed with lipofectamine. The punctate fluorescence in 5d (Now figure 6d) suggests that the DNA is still complexed with lipofectamine, as we discussed in the paper.

We have added the following sentence to the manuscript to clarify this point.

"Signal from SNAP_{Switch} was expected to be generated from oligonucleotides that have dissociated from Lipofectamine in addition to that still complexed but near SNAP-tag."

2.6 Perhaps various transfection reagents could be tested at different concentrations to confirm that the escape phenomenon in Figure 5d is widespread, correlates with transfection efficiency, and not with cell death.

Flow cytometry was used to analyse over 30,000 cells at each time point. Given this, we don't think that cell death is a major cause of the signal that we have observed. However, we have added cell viability data for Lipofectamine 3000 in the supplementary information (SI Figure 10). We have also added additional sentences to the manuscript, as outlined in our responses to points 2.1 and 2.3 above.

2.7 Can the use of a circular plasmid encoding a reporter gene rather than a noncoding oligo be used to correlate between cyto-SNAP and H2A-SNAP positive plasmids and subsequent success/failure in transfection?

Labelling a circular plasmid with any fluorophore is challenging. It has been shown that at >1 fluorophore per plasmid, transfection efficiency is drastically reduced due to decreased ability to dissociate from carriers and impaired transcription (Mol Pharm. 2014;11(5):1359–1368). We agree it would be interesting to investigate the relationship between SNAP_{Switch} activation and expression of a

reporter gene, however this would require optimisation of labelling efficiencies which is something we wish to explore in a future report.

2.8 *There is gathering evidence that Lipofectamine 2000 can evade microtubule-dependent trafficking and therefore the endosomal pathway [Sci Rep 6, 25879 (2016)] and so the claim of endosomal escape may require some qualification.*

The following is quoted from the above reference:

“LFN, by avoiding active transport along the cytoskeleton components, does evade lysosomal degradation, thus in turn increasing the probability of DNA release into the cytosol likely by the formation of multiple transient pores over time within the endosomal membrane”

Although the complexes do not traffic along microtubules, the authors propose they are still contained within endosomes and hence, the term endosomal escape is appropriate.

2.9 *Was the Lipofectamine to DNA ratio optimised as directed in the protocol? Flow data will help greatly here to confirm high transfection with minimal cell death.*

Please refer to our response to point 2.6 above.

2.10 *The opening sentence in para. 3 is potentially confusing. I think perhaps the authors mean fluorescent fusions have traditionally been used to detect and track proteins of interest rather than label subcellular structures.*

We have reworded this sentence as outlined below to be clear that fluorescent fusion proteins are used in the context of both.

“Fluorescent fusion proteins have long been used to label specific locations within cells and to track localization of inbound proteins of interest.¹³”

2.11 *HPLC methods: 0.1% TFA is written twice.*

We have corrected this.

2.12 *More flow cytometry data is required in the supplementary information to illustrate gating and representative results for all experiments.*

We have included example flow cytometry data in SI Figures 15 and 16 to illustrate this.

2.13 *The clause citing Reference 3 does not refer to transport of inbound biomolecules as does the rest of the sentence.*

This has been fixed.

2.14 *Please clarify which Lipofectamine reagent is used in the final paragraph of the introduction section and in para. 2 of the Discussion.*

We have clarified this sentence to be clear that Lipofectamine 3000 was used.

2.15 *Sensor characterization in solution section: You refer to a ‘significant difference’ but no statistics are reported. Perhaps ‘observable difference’ would be less ambiguous here?*

Statistical significance has been added to this section.

2.16 *Discussion, para. 2: You refer to TfR everywhere except here, where CD71 is used. Perhaps this could be more consistent.*

The reference to CD71 has been removed.

2.17 *Figure captions concerning 'normalisation' – the images should have been collected with the same exposure settings and been processed identically making this statement unnecessary. What exactly are the data normalised to?*

All images were taken with the same exposure settings and processed identically. We have reworded the captions to make this clearer.

2.18 *Supp figures 6, 7: Are the images labelled as DIC really DIC? If so, the bias should have been increased to obtain better images as contrast is very low making it almost impossible to distinguish cells from the background.*

The reviewer is correct that these are bright field images. We have corrected the labelling of these images and increased the contrast to help identify the cells

2.19 *Figure 5: What is the timepoint at which the images were taken?*

The time point is 16 hours and has been added to figure caption.

2.20 *Typographical/grammatical corrections:*

Abstract: SNAPswitch

Discussion, para. 1: Missing full stop; 'processes such occurs in'

'The spatial overlap ... is measured by microscopy and statistical analysis (...) is then used'

Supplementary Figure 8 is missing labels (a), (b) and panels seem to be in a different order than the caption suggests.

All of these have been fixed.

Reviewer 3

3.1 *Even though it is a topical area of review, the present tags have much more better resolution and the current study did not show any data to suggest use of this technology for high-throughput analysis. The manuscript in the current form is unsuitable for publication and with utmost respect for the work presented have to unfortunately reject it.*

In the context of this paper, we are referring to flow cytometry as a high-throughput technique to analyse thousands of cells per second, compared to microscopy which is limited to analysing tens to hundreds of cells over a period of hours. We have reworded the manuscript as follows, to be clearer about what we refer to as high-throughput analysis.

“In addition, traditional colocalization analysis is low-throughput, as the number of cells that can be analyzed is limited (typically 10 – 100s of cells over hours) compared to other techniques such as flow cytometry (thousands of cells per second).”

and

“Through synthesis of a quenched and attachable SNAP-tag substrate, we have developed a sensor that enables high-throughput (thousands of cells per sample, measured in less than a minute) and quantitative tracking of biomacromolecules in live cells.”

We disagree with the reviewer that other techniques have a better resolution (see response to point 3.3 below and point 1.1 above)

3.2 *My first issue is that the trafficking of transferrin or transferrin antibody follows a very quick uptake within 5 mins with recycling within 15 mins. The quality of images just does not allow to visualize punctate with high resolution which can be even seen with usual tag. For example, when transferrin internalizes a vesicular structure is observed and within 10 mins a tubulovesicular structure appears indicating recycling endosomes and large vesicular lysosomes can be visualized in 20 mins. Generally, the experiments are pulse chase, starting the incubation at 4 degrees with warming the media and performing imaging. Again, I was not able to visualize any structural minutiae with this tech which can be seen even with just labeled transferrin. The current method does not show any superiority in imaging.*

The significance of our localization assay is not that it improves the resolution of microscope images. The significance of this work is that imaging is not required, and that flow cytometry can be used to determine localization to a resolution that is better than super-resolution microscopy. To demonstrate this, we have included an additional section (Resolving Membrane Orientation of Cargo) that shows how the localization sensor can be used to determine which side of an organelle membrane a SNAP_{Switch} labelled protein is residing (Figure 5).

While synchronising internalisation by incubating at 4^o followed by warming to induce internalisation is a commonly used technique, it has a significant limitations which means we think it should be avoided. The low temperature step inhibits CLIC/GEEC endocytosis for a number of hours, while clathrin-dependent endocytosis recovers rapidly. CD44 is internalised by CLIC/GEEC and TfR is clathrin dependant. If we incubated the cells at 4^o we would be biasing the internalisation and trafficking in favour of the clathrin-dependant pathway.

We have added the following sentence to the manuscript to explain this point.

“Binding of antibodies at low temperature (4 °C) was avoided as anti-CD44 is internalised by the clathrin-independent carrier/GPI-AP enriched early endosomal compartment (CLIC/GEEC) pathway,³⁵ which takes longer to recover than clathrin-mediated endocytosis on return to 37 °C.³⁶”

3.3 The similar issue is with experiments done for endosomal escape. First lipofectamine 3000 based lipoplexes do not form stable nanoparticles and due to very high polydispersity of lipofectamine a lot of aggregation is generally observed on plates. Having said that there have been lot of studies showing trafficking of these complexes and again the image quality was not high enough to pinpoint any differences. The time scales chosen for the events are generally associated with microscopy techniques that generally have low resolution. The state of the art tools have quantified the escape (for siRNA) that has been relatively low and DNA amount is even lower which again was very hard for me to make any visual judgment. There is a cell membrane label, associated with AF488, some cytosolic amounts and little if any label that may not still be attached with the nanoparticle. Again H2A snap images are confusing (especially the membrane localization) So, based on previous publication and quality of images and number of cells imaged to quantify diminishes enthusiasm for the current manuscript

Again, as outlined in our response to point 3.2, we are not claiming the image quality gained from our localization assay is superior to other imaging techniques. We are arguing that microscopy is not required to determine the localization. The images are provided as validation of the flow cytometry data and that the localization assay works.

The reviewer has highlighted one of the major limitations with relying on microscopy images, which is that to determine endosomal escape from microscopy images is very subjective. The use of flow cytometry to quantify signal from thousands of cells removes this subjectivity. We have added the following sentences to clarify this point.

“A critical limitation of using microscopy to pinpoint endosomal escape is the subjectivity of image analysis. Images are often assigned as having a “punctate” (trapped material) or “diffuse appearance” (escaped material).⁷ The use of flow cytometry to quantify signal from thousands of cells removes this subjectivity.”

We thank the reviewer for pointing out that the figure caption of 5d-f is not clear. For the microscopy images the DNA was not dual stained. The AF488 signal in the fluorescence microscopy images is from an AF488 WGA membrane stain, that was used to help visualise material that is inside the cell. Dual staining was only used in the flow cytometry experiments (Figure 5b,c and g) to account for differences in cellular uptake. The figure caption has been revised to clarify this point.

3.4 Finally, my biggest critique is that authors are messaging the work to be useful for highthroughput analysis without providing any data. Furthermore, tools like CLIP or other tools with peptide tags (Proc Natl Acad Sci U S A. 2018 Dec 18;115(51):12961-12966. doi: 10.1073/pnas.1808626115. Epub 2018 Dec 5) and FRET based quenchers have been shown to give high resolution images (Integr Biol

(Camb). 2013 Jan;5(1):224-30. doi: 10.1039/c2ib20155k.) and many others have been shown to be used prior. The idea presented here could have been exciting if it could decipher anything new in terms of trafficking or gave a better quality image that can be interpreted easily.

As discussed above in response to point 1.1, we have revised the manuscript to clarify what we mean about high throughput analysis. Our claim is that using flow cytometry to analyse >10,000 cells is higher throughput than using fluorescence microscopy which typically analyses significantly fewer than 100 cells and takes significantly longer to perform this analysis.

The papers that the reviewer has highlighted are excellent examples of image-based techniques to investigate localization. But they are all limited by the need to perform microscopy, which is inherently slower than the analysis required for flow cytometry. We have included these references in the manuscript and included a more thorough explanation how our localization differs from these techniques as outlined in our response to points 1.2 and 1.4 above.

REVIEWERS' COMMENTS:

Reviewer #1 (Remarks to the Author):

This article only reported a fluorogenic probe for SNAP-tag, and very similar compounds have been reported by nagano, johnsson and others almost 10 years ago. Anyone who had experiences of snap fusion protein expression should know that the expression of the fusion protein is unstable, that is, in the same cell line, some cells can express SNAP in some cells, some express less, some No SNAP fusion protein even appeared. This will result in some different cells in the vicinity of some fluorescent, some not, some strong fluorescence, and some weak fluorescence when stained with fluorescent substrate. Therefore, SNAP technology can only be used for qualitative research, not for quantitative research. The so-called quantitative detection of protein and DNA in this article is not feasible. It must be artificially selected cells that express well to observe, and once the flux test is performed, the method is not feasible. I have mentioned all these concerns in the last comment, but the author did not face these doubts, but vaguely followed. It is recommended that the nat commun editorial department refuse to publish this paper.

Reviewer #2 (Remarks to the Author):

I am satisfied by the new revised version. The authors have incorporated all the recommendations made where appropriate and justified the recommendations made by the reviewers.

The the revised manuscript is acceptable for publication.

Reviewer #3 (Remarks to the Author):

I think the authors have done a good job in addressing reviewer comments. However, it still not clear to me how localization is being observed. Yes, using flow cytometry is quantifying if the delivered molecules is interacting with the stably transfected snap-tag location of interest. Now, if the location of interest is stationary then we can say it has localized to that spot on the cell, however cellular structures are dynamic and there localization from the periphery towards the nucleus shifts and that is something microscopy is beneficial to establish localization. Spatiotemporal localization seems will not be possible with this approach. Flow is used to establish quantification and indirect localization which this method also does. It was difficult to say if this was better than the other methods. Also, nearly all reviewers were confused with the high-throughput language at it pertains to more sample tests than number of cells imaged or counted. One might argue that microscopes like OPERA can do high-throughput imaging (measure 96 well plate in high resolution) and therefore the logic of only flow being able to do this can be made more stronger or modified. I would say though that the authors did give convincing rebuttal and therefore I think I would request rebuttal based on my comments above as I have no additional questions for the experiments.

Reviewer 1

This article only reported a fluorogenic probe for SNAP-tag, and very similar compounds have been reported by nagano, johnsson and others almost 10 years ago.

We think the reviewer 1 still doesn't fully understand the concept behind the SNAP_{Switch} assay, and how it differs from the previous work they refer to. All previous SNAP assays only identify SNAP-tagged protein in cells, they do not measure interactions between the SNAP-tagged proteins and other materials. The compounds synthesised by Nagano and Johnsson only allow for labelling of SNAP-tagged fusion proteins. The innovation in the work presented here is that the SNAP_{Switch} is attached to the DNA/nanoparticles and SNAP-tagged protein activates the SNAP_{Switch} fluorescence and qualitatively determines the interaction.

An analogy for this assay is tracking marathon runners during a race. Runners (DNA/nanoparticles) need to get from point A to point B, but must pass through a series of check points to ensure they haven't taken a short cut.

The traditional way of doing this is to have a series of marker points (SNAP tagged protein labelled with a fluorescent dye) around the course and have a marshal at each marker point marking off each runner's number (fluorescent dye different to the SNAP tagged protein) as they pass through. In this part of the analogy, the marshal is akin to performing colocalization analysis of a nanoparticle/DNA with a cellular marker (e.g. LAMP1 to look at lysosomal association). However, if a marshal is only present for part of the race (say the middle half), they may miss the fastest and the slowest runners. Similarly for co-localisation analysis, unless you acquire and analyse images throughout the entire experiment, interactions will be missed.

The SNAP_{Switch} assay is akin to using a microchip tracker on the runner. The marker points (SNAP-tagged protein) are still present, however you do not need a marshal at each point to check the runner has passed through the check point. You simply scan (analyse using flow cytometry) the microchip (SNAP_{Switch}) at the end of the race.

Anyone who had experiences of snap fusion protein expression should know that the expression of the fusion protein is unstable, that is, in the same cell line, some cells can express SNAP in some cells, some express less, some No SNAP fusion protein even appeared. This will result in some different cells in the vicinity of some fluorescent, some not, some strong fluorescence, and some weak fluorescence when stained with fluorescent substrate. Therefore, SNAP technology can only be used for qualitative research, not for quantitative research. The so-called quantitative detection of protein and DNA in this article is not feasible. It must be artificially selected cells that express well to observe, and once the flux test is performed, the method is not feasible.

The review is correct that transient expression of SNAP fusion proteins can result in variable expression levels within the same cell line. That is why for all the quantitative measurements we generated single cell clones. We also included an antibiotic selection to ensure continued stable selection of the SNAP-tag. To clarify this point we have revised the experimental section of the manuscript.

We established that all cells express similar levels of SNAP-tag by treating the cells with a cell permeable fluorescent SNAP substrate and analysing the cells using flow cytometry. We have included the raw histogram plots in the SI to make this clearer.

Furthermore, the levels of SNAP-tag expressed in the cells are >100 fold higher than the amount of SNAP_{Switch} and cargo delivered to the cell. This means that even if the expression levels in the cell decrease, there remains a large excess of SNAP-tag to activate the SNAP_{Switch}. To continue with the marathon analogy, there are 100 lanes for each runner to choose from when they go through a check point. Even if half of the lanes get closed, there are still many more lanes than runners. Therefore, we do not need to 'artificially' select cells that express high levels of the SNAP tag.

Reviewer 3

It still not clear to me how localization is being observed. Spatiotemporal localization seems will not be possible with this approach.

We can determine if material has localised to a specific organelle (cytosol, nucleus, or endosome), or if it has interacted with a specific receptor (e.g. TfR). The reviewer is correct that this technique does not distinguish between cytosolic material that is close to the nucleus vs close to the plasma membrane. However typically the most important information required for therapeutic delivery is localisation within an organelle, rather than where the organelle is localised inside the cell. To clarify this point, we have amended the discussion to highlight that we are quantifying organelle localisations:

" SNAP_{Switch} is also able to determine localization within organelles at very high resolution"

One might argue that microscopes like OPERA can do high-throughput imaging (measure 96 well plate in high resolution) and therefore the logic of only flow being able to do this can be made more stronger or modified

While microscopes like the Operetta can image samples in a 96 well format, the throughput is still 1-2 orders of magnitude slower than flow cytometry. For example, a flow cytometer analyses the total fluorescent intensity of a cells at up to 20,000 cells per second. The operetta can acquire a few fields of view per second. At high-resolution, each field of view will only contain tens of cells. Additionally, each image is a slice through the cell, so does not measure the entire volume of the cell. Finally, the analysis of co-localisation is subjective and more time consuming than quantifying the SNAP_{Switch} signal using flow cytometry.